# Functional MRI of imprinting memory in awake newborn domestic chicks
Mehdi Behroozi [1,6] ✉, Elena Lorenzi [2,6] ✉, Sepideh Tabrik[3], Martin Tegenthoff[3], Alessandro Gozzi [4], Onur Güntürkün [1,5] & Giorgio Vallortigara [2]

Filial imprinting, a crucial ethological paradigm, provides insights into the neurobiology of early learning and its long-term impact on behaviour. To date, invasive techniques like autoradiography or lesions have been used to study it, limiting the exploration of whole brain networks. Recent advances in fMRI for avian brains now open new windows to explore bird's brain functions at the network level. We developed an fMRI technique for awake, newly hatched chicks, capturing BOLD signal changes during imprinting experiments. While early memory acquisition phases are understood, long-term storage and retrieval remain unclear. Our findings identified potential long-term storage of imprinting memories across a neural network, including the hippocampal formation, the medial striatum, the arcopallium, and the prefrontal-like nidopallium caudolaterale. This paradigm opens up new avenues for exploring the broader landscape of learning and memory in neonatal vertebrates, enhancing our understanding of behaviour and brain networks.

Filial imprinting is a learning process by which the young of some organisms can learn about a conspicuous object, usually the mother or siblings, by simply being exposed to it for a short period of time soon after birth[1]. It owes its great popularity to the work of Nobel-prize-winning ethologist Konrad Lorenz[2], but it was originally described by Douglas Spalding[3] in the offspring of some nidifugous (precocial) bird species, such as chicks or ducklings (see[4]). Visual imprinting has been mostly studied, though acoustic or olfactory imprinting can be observed as well, the latter being prominent in mammals[5].

Although in principle visual imprinting can occur with any kind of object, research has shown that the process is actually assisted by a set of biological predispositions which guides an animal's attention towards those object features that are more likely to be observed in social partners—e.g. preferences in domestic chicks include simple features such red colour (which is prominently observed in the head region of conspecifics), or self-propelled motion (which is typical of living things), as well as more complex assembly of features such as face-like stimuli or biological motion in point-light displays (review in refs. [4,6]). Brain research has shown that biological predispositions are associated with the activation of areas of the so-called *Social Behaviour Network*, and in particular of the lateral septum for motion stimuli and of the nucleus taenia (homologous of the mammalian medial amygdala[7]) for face-like stimuli (review in ref. [6]).

Interest in filial imprinting quickly spanned from behavioural biology to psychological development and psychopathology, inspiring, for instance, John Bowlby's theory of attachment, which postulates a crucial role of the mother-child bond for subsequent psychological development and, complementarily, the psychiatric outcomes associated with early mother deprivation (recent reviews in refs. [8,9]).

In the 70's filial imprinting served as a model for the neurobiological investigation of memory. Gabriel Horn and colleagues (review in ref. [10]) identified an associative brain region involved in the formation of an imprinting memory, the intermediate medial mesopallium (IMM according to the new avian brain nomenclature; previously referred to as IMHV, intermediate medial hyperstriatum ventrale[11,12]). IMM proved to be crucial during the acquisition phase of the visual imprinting memory. More precisely, it was found that exposure to the imprinting object was associated with changes in the left but not in the right IMM[13,14]. Subsequent studies with auditory imprinting revealed that the imprinting-related area extended ventrally into a medialmost nidopallial area, the nidopallium medial pars medialis (NMm)[15,16]. Here we will use the label medial nidopallium/mesopallium (MNM) to jointly label the mesopallial and nidopallial entities of the imprinting area.

Experiments involving sequential lesions, first to one side of IMM and subsequently to the other[17,18], suggested that the store in the left IMM is only

[1]Institute of Cognitive Neuroscience, Department of Biopsychology, Faculty of Psychology, Ruhr University Bochum, Universitätsstraße 150, Bochum, Germany. [2]Center for Mind/Brain Sciences, University of Trento, Piazza Manifattura 1, Rovereto, TN, Italy. [3]Department of Neurology, BG-University Hospital Bergmannsheil, Ruhr-University Bochum, Bürkle-de-la-Camp-Platz 1, Bochum, Germany. [4]Functional neuroimaging laboratory, Istituto Italiano di Tecnologia, Rovereto, Italy. [5]Research Center One Health Ruhr, University Research Alliance Ruhr, Faculty of Psychology, Ruhr University Bochum, Bochum, Germany. [6]These authors contributed equally: Mehdi Behroozi, Elena Lorenzi. ✉e-mail: mehdi.behroozi@ruhr-uni-bochum.de; elena.lorenzi@unitn.it

temporary, and the right IMM is implicated in transferring information from the left IMM to another, unknown brain region dubbed S', and that this transfer appears to be complete within 6 h after the end of exposure[19]. Thus, to cite Gabriel Horn's words *'We are still some ways from being able to visualize, through the microscope or by using brain imaging techniques, the neural trace of (imprinting) memory'*[10]. To overcome the technical limitations, recent advancements in functional magnetic resonance imaging (fMRI) turned it into a cornerstone neuroscientific technique. This powerful, non-invasive procedure serves as an indirect measure of neuronal activity throughout the entire brain, offering a comprehensive perspective at the network level. It appears particularly well-suited to finally find the so-called S', being it a region or a neural network. To this end, we have developed an awake fMRI platform to explore the imprinting network and the long-term store of imprinting memories in newly hatched chicks.

We exposed (imprinted) chicks on either a preferred (red) or a non-preferred (blue) colour. After exposure, awake chicks were tested with a sequence alternating the two colours in the scanner. We could demonstrate that chicks imprinted on red colour showed activity in pallial and subpallial brain regions involved with storage and memory retrieval, such as the medial striatum, the arcopallium, the hippocampus, and the nidopallium caudolaterale (a presumed avian equivalent of mammalian prefrontal cortex). Chicks imprinted on blue showed less activity in the same regions; however, during the last 20 min of scanning when presented with the red, these chicks showed activity in the mesopallium and the *Social Behaviour Network*. The first exposure to the colour red, a predisposed feature for social attachment, thus started a process of secondary imprinting, activating a brain network known to be involved in socially predisposed features at birth. We thus, first, established a reliable platform to investigate the long-term imprinting memory. Second, our results might shed light on the so-called S', the neural basis of the long-term imprinting memory storage which was unknown up to now.

## Results

The present study aimed to better understand the neural networks underlying the different learning stages of filial imprinting: memory acquisition, long-term memory storage, and retrieval. To tackle these ambitious questions, we decided to establish a fully non-invasive awake fMRI protocol for newborn chicks. Using the awake fMRI platform, we were able to capture dynamic neural processes in real-time of imprinting memory at the whole brain level, allowing us to observe and analyse the intricate interplay of brain regions involved in filial imprinting memory without interfering with the natural state of the subjects.

### Establishment of a fully non-invasive and awake fMRI for the chicks

To enable whole-brain fMRI acquisition in awake chicks, we developed a fully non-invasive set-up to minimise head and body movements (Fig. 1).

Before fMRI scans, chicks were imprinted for two days on either a preferred red or a non-preferred blue light ball[20]. Before scanning, chicks were habituated to the scanner noise using a playback of the magnet noise (Fig. 1a). On the third day, after wrapping the animal in a paper tissue to avoid any body-part movements (such as wings and legs), blocks of plasteline were used to comfortably fixate the head, minimising movements and scanner's noise by covering their ears (Fig. 1b).

To record the spontaneous resting-state (to evaluate the stability and reliability of the head fixation system) and task-based BOLD signals, a single-shot multi-slice rapid acquisition with relaxation enhancement (RARE) sequence was adopted from Behroozi et al.[21–23]. Voxel-wise signal-to-noise ratio (SNR) and temporal SNR (tSNR) were calculated over the resting-state (rs-fMRI) and task-based fMRI (tb-fMRI) scans respectively. The tSNR of the RARE sequence in each voxel was calculated after applying motion correction and high-pass temporal filtering (cut-off at 120 s) to remove any linear drift. Temporal SNR in the entire telencephalon ranged from 50 to 100 (Supplementary Fig. 1a, b) for both tb- and rs-fMRI scans. Furthermore, the result indicated highly correlated SNR and tSNR for both rs- and tb-fMRI scans (Supplementary Fig. 1c, d).

In order to verify that adequate fixation was achieved during fMRI scans, we used the realignment parameters and the results of the frame-wise displacement (FD) to evaluate the amount of head motion (Supplementary Fig. 2). Overall, the custom-made restrainer yielded a low level of head movements. There were only 2.02% (218 volumes) and 1.08% (19 volumes) of fMRI volumes with FD higher than 0.2 mm (~40% of voxel size) over all subjects in the task-based and resting-state experiments, respectively (Supplementary Fig. 2A). The median of frame-wise displacement was ~0.03 mm for both tb-fMRI and rs-fMRI experiments. However, most head movements occurred in the y-direction (Supplementary Fig. 2b, c). The respective violin plot information for translations in the y-direction is as follow: tb-fMRI: max/min = 0.22/−0.31 and median ~0; and rs-fMRI: max/min = 0.28/−0.30 and median ~0. The higher motion parameters in the y-direction were most likely due to the design of the head restrainer, which allowed movements in the dorsoventral direction to avoid blocking the throat.

### Distinct BOLD response to identify the acquisition and long-term storage of imprinting memory

We recorded the whole brain BOLD signals from 17 head-restrained awake chicks already imprinted to either a preferred colour, red ($n = 9$), or a non-preferred colour, blue ($n = 8$). During fMRI scanning, animals were presented with both colours (Fig. 1c), which depending on the previous imprinting training could represent either the imprinted or the control colour. The two colours were presented in a block design manner and a pseudo-random order (48 trials, 24 per condition, see Methods). For the preferred colour group, the imprinting colour (Imp) was red and the control

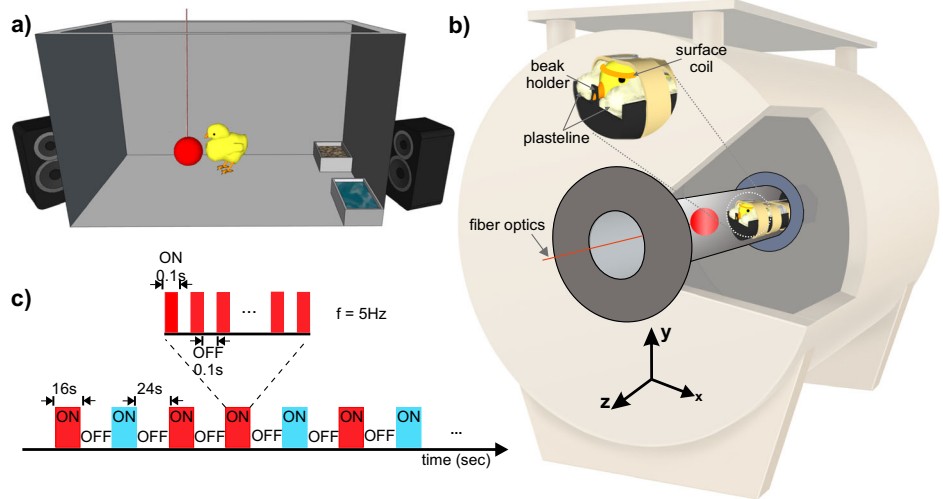

**Fig. 1 | Experimental setups and stimulation sequence for awake chick fMRI. a** Imprinting cage. Newborn chicks were first exposed to a hollow plastic ball with a flickering red/blue light at a frequency of 5 Hz. **b** Custom-made restrainer and 7 T fMRI system. Awake chicks were placed in an MR-compatible tube. To immobilise non-invasively the animals, a beak holder was used to control the beak movements and blocks of plastelines were used to cover the ears and reduce head movements. To avoid body-part movements, animals were wrapped in paper tissue before fixating the head. Subsequently, the animal's body was taped to the restrainer. **c** A sequence of the block design experiment paradigm. Visual stimuli were presented in blocks of 16 s followed by 24 s dark. During the ON blocks, the visual stimulus (red/blue light) flickered at a frequency of 5 Hz. All parts of this figure were created by the authors (M.B., E.L.).

(Cont) was blue, while for the non-preferred colour group the imprinting colour was blue and the control red. To identify the long-term storage of imprinting memory, we first used the contrast of Imp > Cont, where 'Imp > Cont' indicates greater activity for the implicit condition (Imp) compared to the control condition (Cont), by combining both groups in a conventional generalised linear model (GLM) based statistical analysis. The first-level results at the single-subject level were then entered into a second-level analysis (random-effect modelling, $Z = 2.3$ and $p < 0.05$ family-wise error (FWE)) to illustrate the activation clusters at different networks of chick prosencephalon.

Before fMRI scans, chicks were exposed to the imprinting stimulus for 2 days, during which they learned the features of the imprinting object and stored them as a long-term memory[1]. Therefore, we expected to find activation in regions involved in memory retrieval. Surprisingly, $Red_{Imp} + Blue_{Imp} > Blue_{Cont} + Red_{Cont}$ contrast showed no significantly activated cluster in the chick brain. The activation patterns for both contrasts, $Red_{Imp} + Blue_{Imp} >$ baseline and $Blue_{Cont} + Red_{Cont} >$ baseline, were highly similar (Fig. 2a). To get to the bottom of this interference, we examined the interaction between the group factor and the Red vs. Blue contrast by analysing the following contrasts: $Red_{Imp}$ vs. $Blue_{Imp}$, $Blue_{Cont}$ vs. $Red_{Cont}$, $Red_{Imp}$ vs. $Red_{Cont}$, and $Blue_{Imp}$ vs. $Blue_{Cont}$. As illustrated in Fig. 2b and c, robust BOLD activation patterns were found within the telencephalon for the contrasts: $Red_{Imp} > Blue_{Imp}$ (indicating greater activity in $Red_{Imp}$ than $Blue_{Imp}$) and $Blue_{Cont} < Red_{Cont}$ (indicating less activity in $Blue_{Cont}$ than $Red_{Cont}$) contrasts. In addition, the $Red_{Imp}$ vs. $Red_{Cont}$, and $Blue_{Imp}$ vs. $Blue_{Cont}$ contrasts demonstrated no significant differences between the different conditions, same colour serving as the imprinting or control stimulus.

To comprehensively investigate the underlying mechanisms behind this discrepancy, we conducted a meticulously designed behavioural experiment aimed at controlling the influence of colour on the chick's preferred choice. As represented in Fig. 3, we found no significant difference in the colour preference between the two groups (two-tailed independent sample $t$-test: $t_{(22)} = 1.601$, $p = 0.124$, $d = 0.654$; mean ± se Red group: $0.718 ± 0.068$; Blue group: $0.558 ± 0.072$). A significant preference for red was detected in both groups together (two-tailed independent sample $t$-test: $t_{(23)} = 2.683$, $p = 0.013$, $d = 0.548$; $0.638 ± 0.051$). These results confirmed the presence of no significant differences between the Red and the Blue imprinted groups with regard to the preference for the red stimulus[24–27]. These results might support the idea that Blue imprinted chicks exposed to the preferred colour red immediately started a process of secondary imprinting toward it inside the scanner.

To this end, we decided to analyse both groups independently to determine the brain activity pattern during the acquisition and the recall phase of a long-term memory of imprinting. While Imp > Cont contrast in the red group showed robust activation clusters in many telencephalic as well as diencephalic regions, in the blue group showed no significant activation clusters.

As shown in Figs. 2, 3, Supplementary Figs. 3, and 4, this is due to chicks' preference for red over blue (as demonstrated through the behavioural experiment), therefore we used the Imp > Cont contrast (blue > red colour) during the first 10 min and Cont > Imp contrast (red > blue colour) during the last 20 min of scanning, to investigate the memory retrieval and memory formation phase of a new imprinting process[28] elicited by the presence of the preferred colour red. The results of the first 10 min scans indicate weak activity in the intermediate medial mesopallium (IMM), the hippocampus (Hp), the medial Striatum (MSt), the nidopallium caudolaterale (NCL) and the n. Taeniae of the Amygdala (TnA) when chicks imprinted with blue colour were initially presented with blue ($n = 8$, $Z = 1.9$ and $p < 0.05$ FEW corrected at the cluster level). However, as illustrated in Fig. 4, the voxel-based group analysis during the last 20 min scans showed robust BOLD responses in different visual prosencephalic regions: the n. geniculatus lateralis pars dorsalis (GLd, which receives direct input from the retina[29]), the right intermediate hyperpallium apicale (IHA, which primarily receives visual thalamic input[30]), the right hyperpallium intercalatum (HI)

and right hyperpallium densocellulare (HD), and bilaterally the hyperpallium apicale (HA, together with HD associative hubs of the thalamofugal pathway[30,31]) of the thalamofugal pathway, bilaterally the n. rotundus (Rot, which is the primary thalamic input region of the tectofugal pathway). Also, parts of the auditory system were activated: bilaterally the ventromedial part of the Field-L complex and the right n. ovoidalis (OV), a thalamic auditory nucleus receiving direct input from the avian homologue of the inferior colliculus (*torus semicircularis*[32]) that projects to Field-L. We detected significant activation clusters in the associative pallial regions nidopallium medial pars medialis (NMm) and bilaterally in the caudal intermediate medial mesopallium (IMM). Within the two interconnected *Social Behaviour Network* and *Mesolimbic Reward System*, we detected a significant BOLD increase rightward in the bed nucleus of the stria terminalis (BNST), the n. accumbens (Ac) and the medial striatum (MSt), bilaterally in the septum and leftward in the posterior pallial amygdala (PoA) and the ventromedial part of hippocampus (Hp-VM).

As illustrated in Fig. 5 and Supplementary Fig. 5, the voxel-based group analysis during the imprinting memory retrieval phase in the red group showed robust BOLD responses in different visual prosencephalic regions: the right GLd, bilaterally in IHA, HI, HD and HA. We found also a significant BOLD rightward increase in the part of the auditory system, OV. Furthermore, we detected a significant increase in the BOLD signal in the associative right MNM (IMM + NMm) and nidopallium caudolaterale (NCL) and in left portions of the caudal mesopallium dorsale (MD) and nidopallium caudocentrale (NCC; all interconnected regions[33–35]). Within the two interconnected *Social Behaviour Network* and *Mesolimbic Reward System*, we detected significant bilateral activation in the ventromedial part of the hippocampus (Hp-VM), while rightward activation clusters in the bed nucleus of the stria terminalis (BNST), in the nucleus accumbens (Ac), in the medial striatum (MSt), in the medial and dorsal arcopallium (respectively AM and AD), in the posterior pallial amygdala (PoA) and in the preoptic, anterior and ventromedial areas of the hypothalamus (respectively POA, AH, and VMH).

## Discussion

Imprinting, a well-known form of early learning, has been widely used in the 70's as a model to study the neurobiology of memory formation (reviews in refs. 10,36). Evidence for a crucial role played by the intermediate medial mesopallium (IMM) and NMm (jointly labelled as MNM) during the acquisition of imprinting memory was obtained. Further studies showed that the store in the IMM is only temporary and that a transfer of information to another, unknown brain region, dubbed S'[37], occurs after approximately 6 h. These studies were conducted with either autoradiographic or lesion techniques and were unable to discover the full imprinting network[38]. To overcome the limitations of traditional methods, fMRI represents a significant leap forward in our ability to investigate and comprehend brain activities. By providing an indirect measurement, BOLD, of the whole brain in various circumstances, this cutting-edge technology offers scientists with a powerful tool for unravelling complex brain networks and sheds light on their roles in diverse cognitive processes, especially memory.

fMRI has been used in several studies to investigate the connection between connectome variations and memory. A recent study in songbirds highlights fMRI's role in tracking song memory development in zebra finch's post-tutorial sessions[39]. The study revealed permanent neural activity changes in auditory perception and song learning, highlighting early sensory experiences. Gazzaley and Nobre[40] explored the neural basis of working memory encoding and maintenance using fMRI. Rahm et al.[41] characterized the neural basis of visual working memory recognition using fMRI by varying recognition needs and similarity between probe items and memory contents. Yang et al.[42] proposed an enhanced connectome-based predictive modelling approach, which showed strong applicability across different cognitive processes and could predict working memory performance in healthy individuals. These findings underscore the potential of fMRI in understanding brain processes that underpin cognitive abilities.

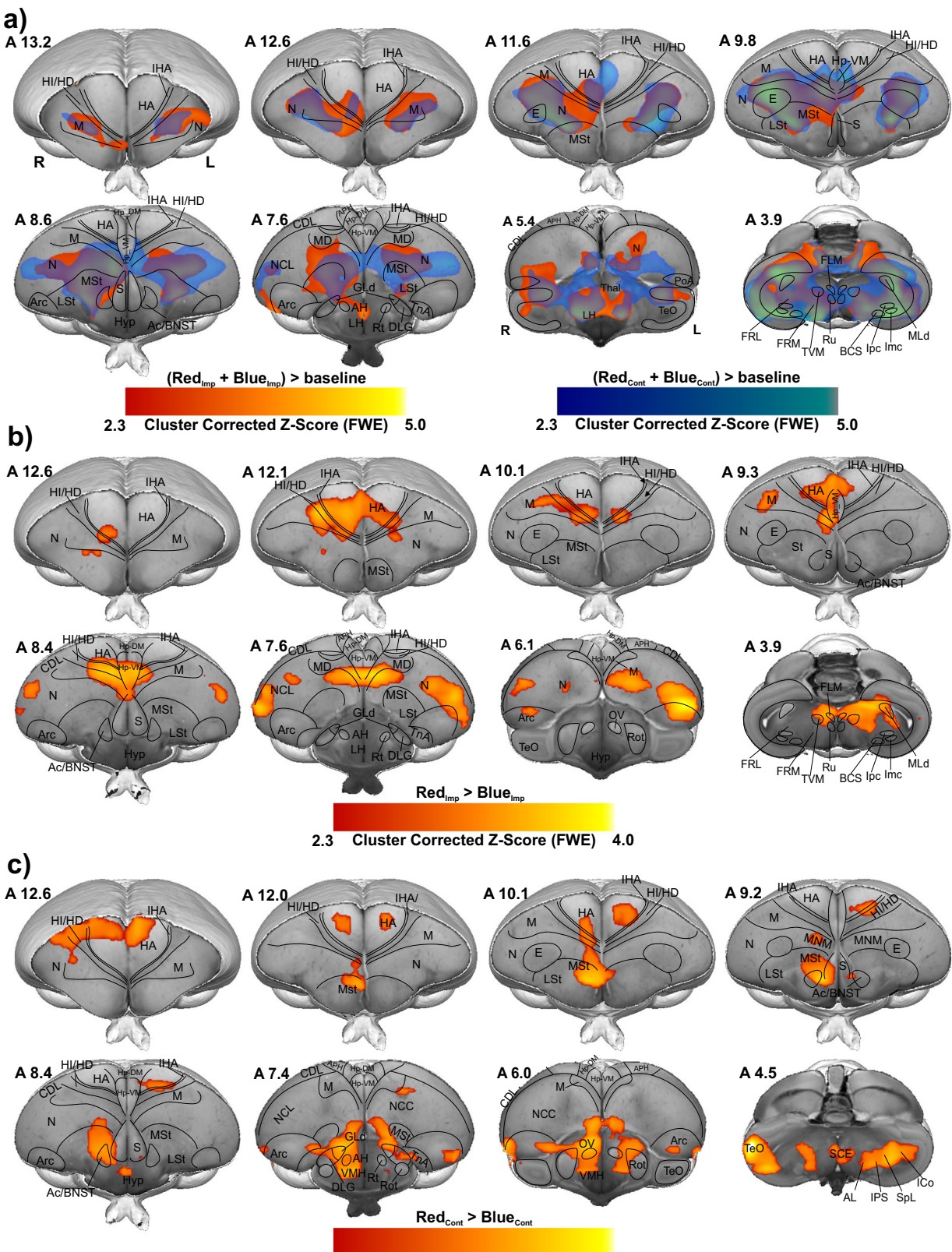

**Fig. 2 | Colour preference of different groups. a** GLM analysis was used to demonstrate activated networks during imprinting and control trials by examining RedImp + BlueImp > baseline (red map) and BlueCont + RedCont > baseline (blue map) contrasts. The colour maps show the activation significance of group-averaged data from 17 chicks (9 red groups + 8 blue groups) in the axial view (group analysis using a mixed model FLAME 1 + 2 method, Z = 2.3, and p < 0.05 FEW corrected at the cluster level). **b** the contrast map shows the significant increase of BOLD signal

during Red colour as imprinting stimulus compared to Blue colour as imprinting stimulus (RedImp > BlueImp contrast, 9 chicks for red group and 8 chicks for blue group). **c** Activation map showing the strong BOLD response during the Red colour as control stimulus for the blue group compared to the Blue colour as control stimulus for red group (BlueCont < RedCont, 9 chicks for red group and 8 chicks for blue group). The functional maps were superimposed on the high-resolution anatomical data at the different levels of an ex vivo chick brain (in grayscale).

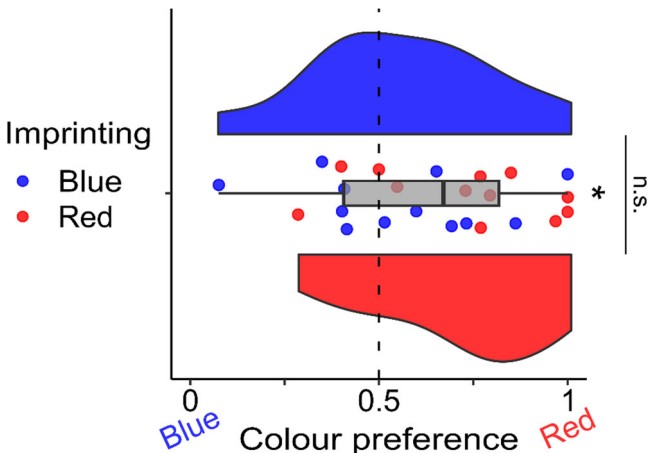

**Fig. 3 | Colour preference after imprinting.** The boxplot in grey represents the colour preference in both red and blue imprinted groups together ($n = 12$ per group). The asterisk represents a significant difference from chance (dotted line). To best represent the data, we provided each subject preference (red points are Red imprinted chicks and blue points are Blue imprinted chicks) and a violin plot for each imprinting group (blue and red respectively) representing the group distribution. No significant difference was detected between the two groups in the colour preference. The data for individual chicks are available in Supplementary Table 1.

Here we established a new non-invasive fMRI protocol to study awake brain activity in newly hatched domestic chicks in order to discover the neural pathways of imprinting and the identity of S'. After two days of imprinting training, with either a preferred (red) or a non-preferred (blue) colour, chicks were exposed to a sequence of the two stimulus colours inside the scanner. In Red imprinted chicks we found a network of brain regions probably associated with the long-term encoding and retrieval of imprinting memory. Interestingly, Blue imprinted chicks did not show such strong brain activity in these brain regions. To further explore the difference between the two groups, we conducted separate analyses for the initial 10 min and the final 20 min of the scanning for the Blue imprinted chicks. The analysis of the first 10 min unveiled that blue imprinted chicks when presented with blue did show an activation (albeit weaker) of the very same brain regions observed when red imprinted chicks were presented with red (Supplementary Fig. 3). Interestingly, during the last 20 minutes of scanning blue imprinted chicks showed a progressively increasing activity when presented with red in brain regions that we know from previous literature are associated with the very first phases of imprinting learning[1,10–12,15–18]. We interpret these findings as follows: when the red colour is available, a new imprinting process begins toward it, as red is highly preferred by chicks (Fig. 4). Such a phenomenon could be (i) a secondary imprinting process starting or (ii) the start of the first imprinting on red, given that the initial imprinting with blue was notably weaker or absent. In the following sections, we will refer to this as the acquisition phase of new imprinting.

Visual information reaches the pallium both via the tecto- and the thalamofugal visual pathways. We observed a partial involvement of the nucleus rotundus (Rot), the thalamic link of the tectofugal pathway during acquisition (Fig. 6a). A rotundal involvement had already been reported in imprinted chicks[43] and together with the present results, it could suggests a minor tectofugal role during the early stages of imprinting learning. In contrast, the thalamofugal visual system seems to play a crucial role in processing imprinting information (as also reviewed in ref. 44). This pathway consists of the retinorecipient GLd[33] that projects to the interstitial nucleus of the hyperpallium apicale (IHA) of the visual Wulst, from where secondary projections reach the three pseudo-layers of the Wulst hyperpallium densocellulare (HD), hyperpallium intercalatum (HI), and hyperpallium apicale (HA)[45]. We discovered both during memory formation and retrieval (Fig. 6b) significant activity patterns of all these thalamofugal

components. Indeed, HD of dark-reared chicks exhibits topographically organised responses for red and blue objects[46]. After imprinting on either one of the two colours, such organisation changes along the rostro-caudal axis showing imprinting-related plasticity already in the Wulst.

Previous studies showed that Wulst lesions lead to anterograde amnesia of visual imprinting memory[46]. This possibly results from the loss of visual projections from HD to IMM[47,48], the associative medial pallial area that is crucial for the acquisition of imprinting memory[10]. IMM projects back to HA, establishing a loop[33]. IMM, the ventrally located NMm and the nidopallium caudolaterale (NCL) have been shown to be involved during visual as well as auditory filial imprinting[15,49]. Here we report a significant brain activation in IMM, NMm, and NCL during memory retrieval and, to a much lesser extent, in IMM and NMm during memory formation. Indeed, NMm and NCL undergo long-lasting synaptic changes after multimodal (visuo-auditory) imprinting training[10,49,50]. Imprinting training also impacts cell proliferation in NMm and NCL, but not in IMM[51]. Thus, these three areas play important but differential roles in multimodal filial imprinting learning and the subsequent formation of long-term memory. Note that in the present study, chicks were also exposed to the noise produced by the scanner. Thus, NMm and NCL on the one and the auditory n. ovoidalis (OV)—Field-L pathway on the other side, could conceivably constitute the neural basis for the acoustic component of acquiring (blue group) or retrieving imprinting memory (red group).

However, the interconnected higher associative regions, NMm and NCL do not only play a role for long-term memory-related mechanisms[22,50,52,53]. NMm is also involved in sensorimotor learning and sequential behaviour[54], while NCL, largely accepted as a prefrontal-like field[35], is involved in working memory[55,56], executive control[57,58] and in merging multi-sensory information in long-term memory engrams[59]. This evidence together with the present findings further supports the involvement of these regions in the long-term storage and flexible retrieval of a multimodal imprinting memory trace.

The motor output component of NMm and NCL is established by their projections to arcopallium and medial striatum (MSt)[52,60–63]. Possibly, the initially partially processed imprinting trace is thereby transferred into a striatum-dependent response strategy. As a result, striatal S-R associations are formed and once acquired, drive animal's imprinting behaviour[63]. This also has been shown for passive avoidance learning. Here, the mnemonic nature of MSt (previously lobus paraolfactorius[12]) goes hand in hand with that of IMM[36,64,65], with increased density of synapses and dendritic spines being detectable some days after training in MSt, but not in IMM[65–67]. Additionally, after imprinting training, glutamate receptor binding affinity increases both in MSt and arcopallium[68–70], while, pre-imprinting arcopallial lesions impair memory acquisition[71].

We found enhanced brain activity in the most medial part of MSt both during acquisition and retrieval of imprinting memory, while for the dorsal and medial portions of arcopallium this was only observed for retrieval. These portions of MSt and arcopallium are enriched in the limbic system-associated membrane protein (LAMP)[72]. We also found a strong meso-limbic involvement in imprinting memory in the two interconnected *Social Behaviour Network* and *Mesolimbic Reward System*[73–75]. Here septum was involved only during memory formation. Arcopallium, preoptic area, anterior and ventromedial hypothalamus (POA, AH, VMH) were involved only during memory retrieval. In contrast, Hp, MSt, bed n. of the stria terminalis (BNST), n. accumbens (Ac) and posterior pallial amygdala (PoA) were involved during both memory formation and retrieval. While involvement of these systems in social predispositions associated with imprinting had already been observed[6,76,77], this is the first evidence for their involvement during imprinting. Such involvement could represent the motivational component linked to the association. Indeed, in the context of filial imprinting, emotional-motivational engagement must be particularly pronounced at different stages of the learning process. The septum seems to be preferentially involved during the first stages of imprinting and probably driving the chick's attention toward salient predisposed moving stimuli. Previous studies also revealed septal involvement during the first exposure

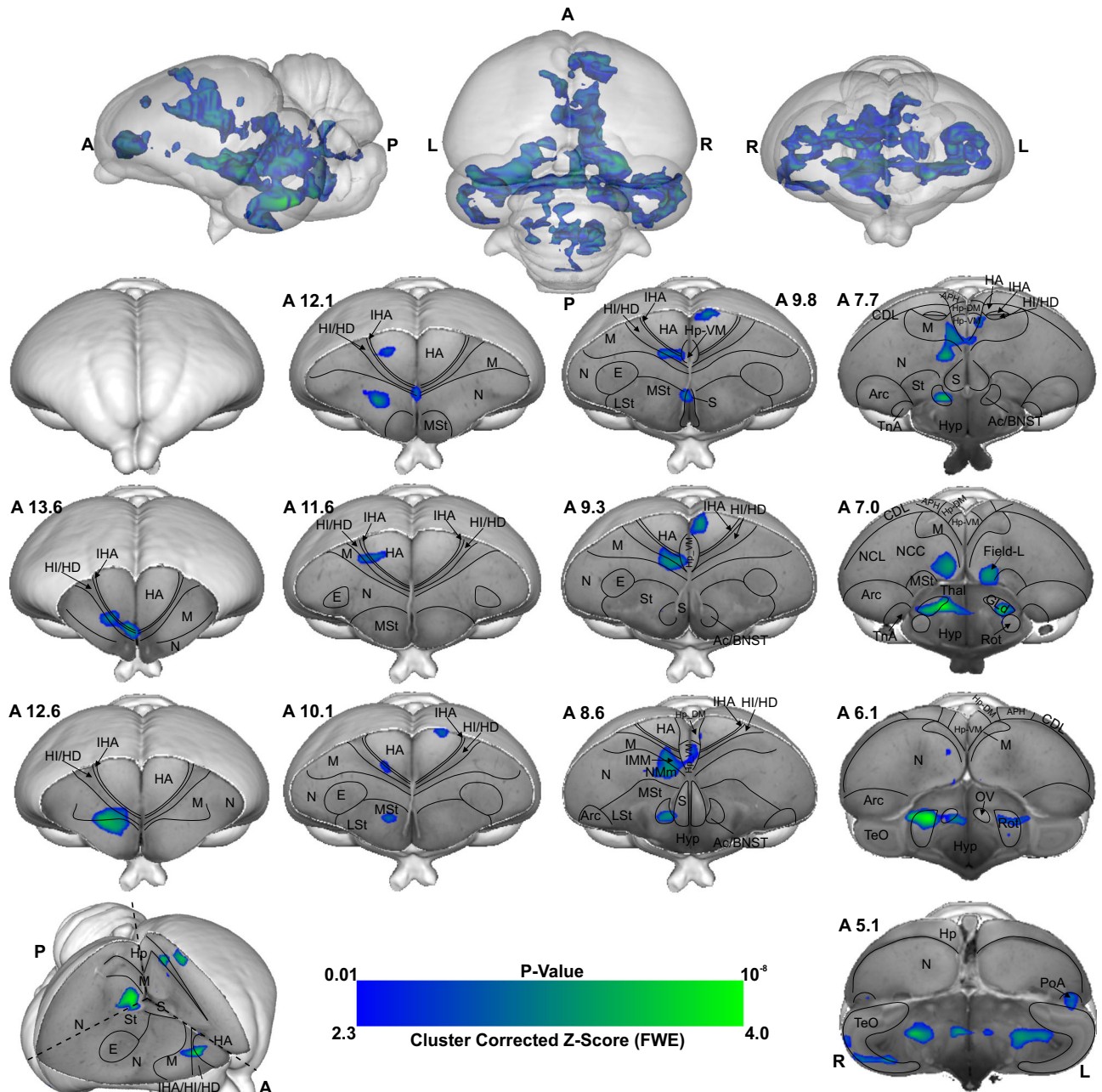

**Fig. 4 | BOLD response pattern during the acquisition of imprinting memory.** Statistical maps for the BOLD signal increase in the contrast of red light versus blue light in the Blue group ($n = 8$, $Z = 2.3$, and $p < 0.05$ FEW corrected at the cluster level). The top row images show the 3D representation of the activation pattern inside a translucent chick brain. A 3D depiction of the chick brain is represented at the bottom of the left column with an example window at the level of A 7.0. Anatomical borders (black lines) are based on the contrast difference of ex-vivo chick brain and Chick atlas[101,102]. The corresponding abbreviations of ROIs are listed in the Supplementary Table 2.

to a red object moving with abrupt changes of speed or an alive conspecific[76,77]. Although BNST, Ac, MSt, and PoA seem to participate in both imprinting memory formation and retrieval, we found greater activity in the red group. Such enhanced activity may suggest a stronger emotional-motivational component after memory consolidation of the imprinting engram.

The HD of the Wulst has bidirectional connections with PoA and Hp[78,79]. We found a hippocampal (Hp) involvement both during imprinting memory formation and retrieval. The hippocampal formation is known for its role in memory in birds and mammals[80]. However, *c-fos* immunoreactivity in chicks revealed also a social role of Hp. The dorso- and ventromedial portions are involved in individual recognition in chicks[81]. The

same portions here were found to be involved in imprinting memory, strengthening a regional specialisation of hippocampus dedicated to social memory functions. Indeed, Hp projects ipsi- and contralaterally to IMM[66] and is involved in filial imprinting[68]. We found a left Hp involvement during filial imprinting memory formation (blue group) and a bilateral one during memory retrieval (red group).

Interestingly, the brain activity pattern was predominantly right lateralised. Among the exceptions was a left Hp involvement during imprinting memory formation (blue group), and a bilateral Hp involvement during memory retrieval (red group). Lateralisation is a common feature in the avian brain, especially at different stages of memory formation[82–85]. Right lateralisation during memory formation has been reported for passive

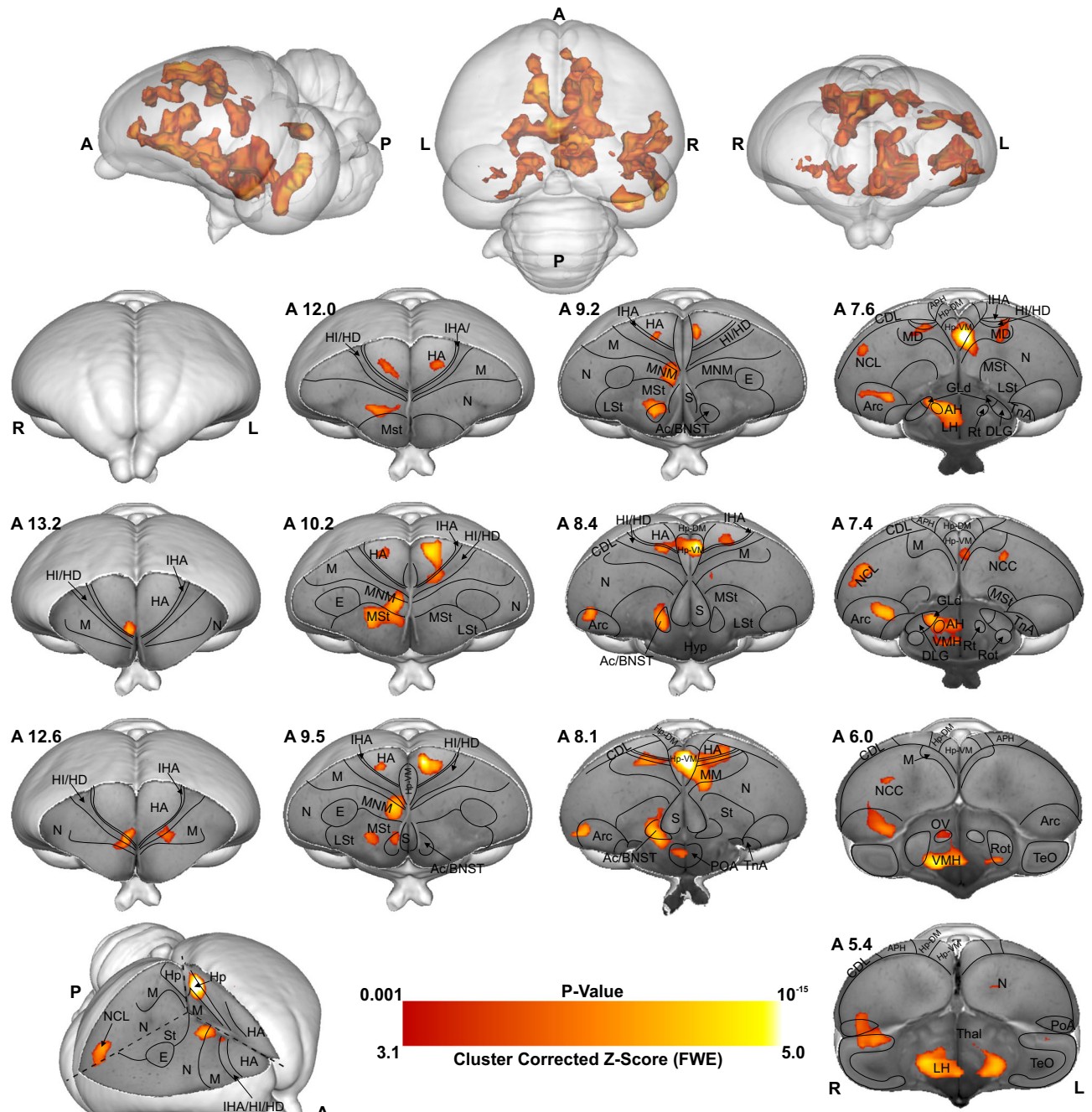

**Fig. 5 | BOLD response pattern during imprinting memory retrieval.** The high-resolution coronal slices at the different levels of an ex-vivo chick brain are in greyscale, while the contrast map represents the activation pattern during the presentation of the preferred imprinting object after imprinting learning has already occurred (Red group, the contrast of red light versus blue light conditions, $n = 9$, $Z = 3.1$ and $p < 0.05$ FEW corrected at the cluster level). The top row images show the 3D representation of the activation pattern inside a translucent chick brain. A 3D depiction of the chick brain is represented bottom left with an example window at the level of A 7.4. Anatomical borders (black lines) are based on the contrast difference of ex-vivo chick brain and Chick atlas[101,102]. The corresponding abbreviations of ROIs are listed in Supplementary Table 2.

avoidance learning[86]. Instead, for imprinting learning, time-shifts have been observed in the lateralisation pattern of IMM. The left IMM is involved at first in learning the features of the imprinting object, while the right IMM dominates during memory consolidation and the subsequent establishment of the long-term storage S'[18,87]. A similar pattern of lateralisation has been proposed in the hemispheric encoding/retrieval asymmetry model (HERA) in humans, where the left hemisphere plays a dominant role during memory encoding and the right during retrieval[84]. Such evidence together leads to the hypothesis of a dual memory system for imprinting, in which different processes—acquisition and consolidation—take place in different hemispheres, with prominent right lateralisation for consolidation processes[18]. Indeed, during memory consolidation, a glutamate injection into the right IMM disrupts imprinting memory, but it does not when injected into the left hemisphere[88]. Our results may add a novel view on the idea of the dual memory system: while the visual thalamofugal nucleus GLd was bilaterally activated during acquisition, only the right side was active during retrieval. It is conceivable that right hemispheric memory consolidation increased top-down projections onto right-sided sensory thalamic nuclei in order to focus attention on learned object properties[89]. This then could activate and synchronize right hemispheric pallial areas

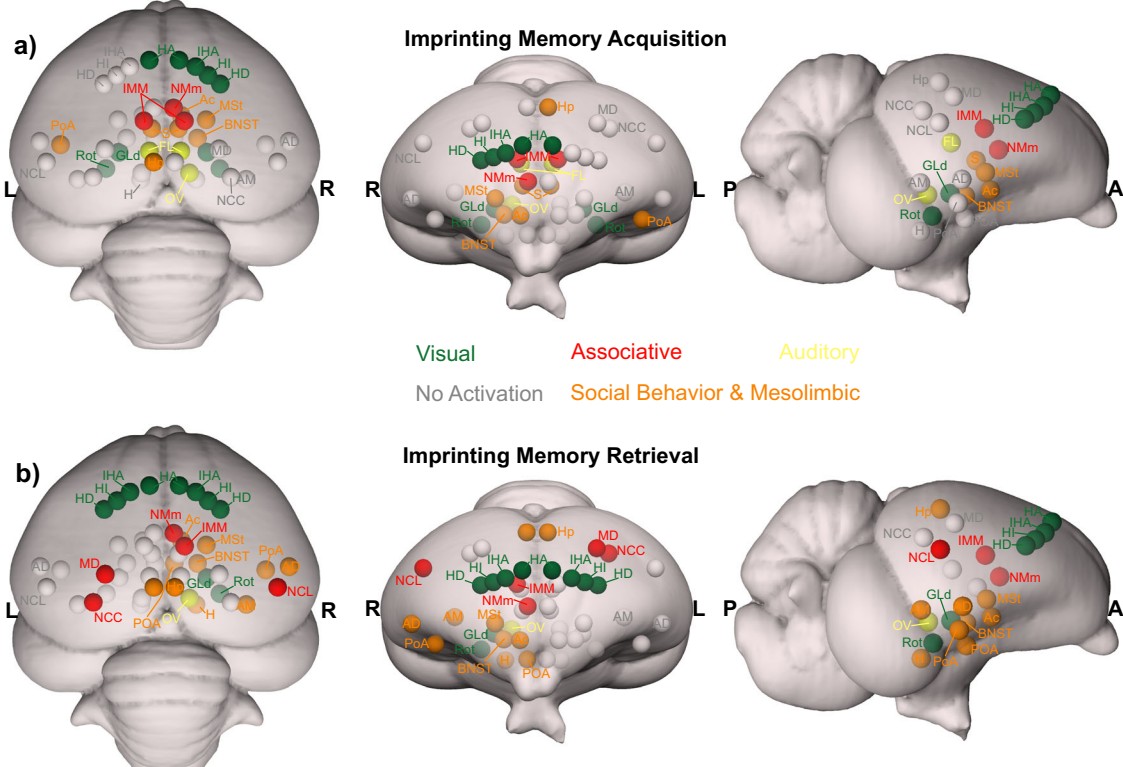

**Fig. 6 | Schematic depiction of the activated prosencephalic areas during different phases of imprinting memory. a** Network activated during imprinting memory acquisition is represented in colourful circles. **b** Network activated during imprinting memory retrieval is represented in colourful circles. The grey circles represent no activation. The corresponding abbreviations of ROIs are listed in the Supplementary Table 2.

according to attentional allocation, thereby inducing a right hemispheric superiority in imprinting memory retrieval[90].

Our findings provide a completely non-invasive paradigm for studying neural mechanisms at birth in newly hatched chicks. Additionally, our data suggests a prosencephalic neural network that, among others, involves the *Social Behaviour Network*, the *Mesolimbic Reward System*, and the medial meso-/nidopallium for long-term storage and retrieval of filial imprinting memory. As to be expected, the networks that could be involved in memory formation and retrieval partially overlapped. However, network activity was more pronounced and further involved arcopallium and NCL in the retrieval condition. Thus, consolidation of imprinting memory seems to result in a strengthening and expansion of the neural system that holds the engram in distributed manner. Within this perspective, the long-searched site for imprinting memory dubbed as S' by Gabriel Horn[91] is possibly this whole network within which the "prefrontal" NCL could be a central hub.

## Methods
### Subjects
All procedures here presented followed all the applicable European Union and Italian laws, and guidelines for animals' care and use and were approved by the Ethical Committee of the University of Trento OPBA and by the Italian Health Ministry (permit number 738/2019). Fifty-three females were used in the present study. Twenty-nine for the MRI procedure: red group (*n* = 10) imprinted to red colour, blue group (*n* = 10) imprinted to blue colour, and resting-state group (*n* = 9). After checking the motion parameters in the fMRI scans, we excluded data from three animals, resulting in *n* = 9 for the red group and *n* = 8 for the blue group (see below). Twenty-four for the behavioural experiment: red group (*n* = 12) imprinted to the red colour, blue group (*n* = 12) imprinted to the blue colour. Each chick underwent the experimental procedure only once. We have complied with all relevant ethical regulations for animal use.

A local commercial hatchery (Azienda Agricola Crescenti, Brescia, Italy) provided fertilised eggs of the Aviagen Ross 308 strain (*Gallus gallus*

*domesticus*). Eggs were incubated and hatched in the laboratory under controlled temperature (37.7 °C) and humidity (60%) in darkness using FIEM MG140/200 Rural LCD EVO incubators. Soon after hatching, chicks were sexed by feather dimorphism, with a black cap on the head in order to prevent any visual stimulation. Twenty-six females were used in the present study. Females were used because they are known to exhibit stronger filial attachment with the imprinting object[92–94]. Each chick underwent the experimental procedure only once. At the end of the experimental procedure, on post-hatching day 3, chicks were caged in groups with water and food ad libitum, at constant temperature (32.3 °C) and with a 12:12 day-night light cycle until they were donated to local farmers.

### Imprinting
On the day of hatching, chicks were caged individually at a constant temperature of 32.3 °C with water and food. In each cage (28 × 40 × 32 cm) the imprinting stimulus, a hollow plastic ball (diameter 3.5 cm), was suspended in the middle (7 cm from the floor, Fig. 1a). Two optical fibres (diameter of 2 mm) inserted in the ball were flickering at 5 Hz. Chicks prefer to imprint on a flickering than on a stationary light[95]. For one group of chicks, the ball was flickering with red light (N = 9, dominant wavelength = 642 nm, intensity = 16.45 cd/m², ), for the other group with blue light (N = 8, dominant wavelength = 465 nm, intensity = 16.45 cd/m²). Being the only light provided in the environment, the established setup by Behroozi et al. [22] and a custom-written MATLAB code was used to automatically switch on and off the light, following a day-night cycle 12:12 (see ref. 96). During the daytime, to habituate the subjects to the noise of the scanner, a recording of the sound was provided twice per day, for a total amount of 5 h per day, by two loudspeakers (Logitech) placed outside the cages.

### Acquisition and pre-processing of fMRI data
All MRI experiments were recorded using a horizontal-bore small animal MRI scanner (7.0 T Bruker BioSpin, Ettlingen, Germany) equipped with a BGA-9 gradient set (380 mT/m, max. linear slew rate 3420 T/m/s). A 72 mm

transmit birdcage resonator was used for radio-frequency transmission. To reduce the motion artefacts resulting from body parts' movements, a single-loop 20 mm surface coil was placed around the chicks' head for signal reception.

**Localiser.** At the beginning of each scanning session, a set of scout images (coronal, horizontal, and sagittal scans) were recorded as localisers to identify the position and orientation of the chick's brain inside the MRI machine. The scout images were acquired using a multi-slice rapid acquisition (RARE) sequence with the following parameters: repetition time (TR) = 3000 ms, effective echo time ($TE_{eff}$) = 41.2 ms, RARE factor = 32, N_average = 2, acquisition matrix = 128 × 128, the field of view (FoV) = 20 × 20 mm, spatial resolution = 0.156 × 0.156 $mm^2$, slice thickness = 1 mm, number of slices = 8, slice orientation = coronal/horizontal/sagittal, with a total scan time of 18 s. This information has been used to position 9 coronal slices in a way (~40° regarding coronal direction) to cover the entire telencephalon to record the fMRI time series.

**fMRI (task).** The blood-oxygen-level-dependent (BOLD) time series were recorded using a single-shot multi-slice RARE sequence adopted from Behroozi et al. [22,97] with the following parameters: TR/$TE_{eff}$ = 4000/51.04 ms, RARE factor = 42, acquisition matrix = 64 × 64, FoV = 30 × 30 $mm^2$, 9 coronal slices no gap between slices, slice thickness = 1 mm, slice order = interleaved. Since the eyes' size is comparable to brain's one, two saturation slices were manually positioned on the eyes to saturate the possible eye movement artefacts, which can corrupt the BOLD signal. A total of 540 volumes were recorded for each animal.

**fMRI (Rest).** Whole-brain resting-state fMRI data (200 volumes) of nine chicks were recorded using a single-shot RARE sequence with the same parameter as the task fMRI sequence.

**Structural MRI.** High-resolution anatomical images were acquired using a RARE sequence with following parameters: TR/$TE_{eff}$ = 6000/42.04 ms, RARE factor = 16, N_Average = 4, acquisition matrix = 160 × 160, FoV = 30 × 30 $mm^2$, 39 coronal slices with no gap between slices, slice thickness = 0.33 mm, total scan time = 4 min.

**Experimental task.** Inside the fMRI machine, chicks were presented with two different stimulus types, imprinted (red/blue) and control colour (blue/red) with the same wavelength and intensity as the training phase. The light stimuli were generated using the established setup by Behroozi et al. [22]. Stimuli were presented in a pseudo-random order in an ON/OFF block design experiment (maximum two trials in a row were of the same colour). The duration of ON blocks was 16 s. ON blocks were interleaved with a rest period of 24 s (OFF blocks, inter-trial interval (ITI)). In total, 48 trials were recorded during an fMRI session from each animal (24 trials per stimulus).

**Apparatus.** A critical issue during awake fMRI scanning of animals is motion artefacts. Therefore, immobilisation of the animal's head is essential to acquire an accurate fMRI time series. To this end, awake chicks were immobilised in a nonmagnetic custom-made restrainer, composed of a beak holder, blocks of plasticine around the head to immobilise it in a comfortable way, and a round RF coil on top of the head (Fig. 1b). Before the head fixation, the animal's body was wrapped in paper tissue to prevent the other body parts' movement (such as wings and feet) to avoid any possible motion artefacts. The animal's body inside the paper tissue was tapped to the main body of the restrainer using a piece of medical tape.

**fMRI data processing**
All BOLD time series were pre-processed using the FMRIB Software Library (FSL, version 6.0.4, https://fsl.fmrib.ox.ac.uk/fsl/fslwiki), the Analysis of

Functional NeuroImages (AFNI, version 20.0.09 https://afni.nimh.nih.gov/), and Advanced Normalization Tools (ANTs, http://stnava.github.io/ANTs/) software. We performed the following pre-processing steps for each run: (i) converting dicom files to nifti format (using dcm2niix function); (ii) upscaling the voxel size by a factor of 10 (using AFNI's 3drefit); (iii) discarding the first 5 volumes to ensure longitudinal magnetization reached steady state; (iv) motion correction using MCFLIRT (which aligns each volume to the middle volume of each run); (v) slice time correction to account for the long whole-brain acquisition time (4000 ms); interleaved acquisitions); (vi) despiking using 3dDespike algorithm in AFNI; (vii) removing non-brain tissue (using BET and manual cleaning); (viii) spatial smoothing with FWHM = 8 mm (using FSL's SUSAN, after upscaling voxel size by factor of 10); (ix) global intensity normalization with grand mean = 10,000 across scanning sessions for group analysis; (x) high-pass temporal filtering to remove slow drifts (cut-off at 100 s); (xi) anatomical brain extraction (using BET function and cleaned manually); (xii) registration of the functional data to the high-resolution structural images using affine linear registration (FLIRT function, six degrees of freedom). For spatial normalization, a population-based template was constructed using antsMultivariateTemplateConstruction.sh script (ANTs). FMRIB's Non-linear Image Registration Tool (FNRIT)[98] was used to spatially normalize the single subject anatomical images to the population-based template as a standard space. The head motion of animals was quantified using framewise displacement (FD)[99]. Three animals' data were excluded due to the excessive head motion (over 20% of volumes were contaminated with FD > 0.2 mm, one from the red group and 2 from the blue group). For the remaining animals, the detected motion outliers were modelled as confound regressors during the general linear model (GLM) analysis to reduce the impact of head motion.

**General linear model (GLM) analysis**
Whole-brain statistical analysis was performed using the FEAT (FMRI Expert Analysis Tool) to assess stimulus-evoked activation patterns. Single-subject GLM analysis was carried out to convolve the established double-gamma avian hemodynamic response function in pigeon brain by Behroozi et al. [22] (the closest brain in the structural organization to the chick brain) to the explanatory variables (on/off stimulation). In the first GLM, we incorporated the complete fMRI timeseries using the following two explanatory variables (EVs) and their temporal derivatives: (i) imprinting trials (indicated by red/blue, 24 trials) and (ii) control trials (indicated by blue/red, 24 trials). In the second GLM, we employed three EVs and their temporal derivatives: (i) imprinting trials (last 16 trials); (ii) control trials (last 16 trials); (iii) junk trials (the first 16 trials comprising 8 imprinting and 8 control trials, were used as habituation period to the real magnet environment). In addition, six estimated head motion parameters (three translations and three rotations) and outlier volumes detected based on the FD analysis were modelled as confound EVs to remove the residual motion artefacts.

**Visualization**
To visualize the results, we took advantage of the high-resolution anatomical image acquired for another study. Briefly, five post-mortem chick brains were scanned using a fast-low angle shot (FLASH) sequence with following parameters: TR/TE = 50/4 ms, N_average = 6, acquisition matrix = 400 × 400 × 500, voxels size = 0.05 × 0.05 × 0.05 $mm^3$, total scan time = 19 h 48 min. The population-based template was co-registered nonlinearly (using FNIRT) to the high-resolution anatomical image of the chick brain. The contrasts of interest, eventually, were non-linearly warped to the high-resolution anatomical image. MANGO software (http://ric.uthscsa.edu/mango/mango.html, version 4.1) was used for 3D visualization of the activation patterns. Surf Ice software (https://www.nitrc.org/projects/surfice/, version v1.0.20201102 64 bit x86-64 Windows) was used for surface rendering the chick brains with overlays to illustrate activated networks during imprinting acquisition and retrieval memory.

## Behavioural experiment

Similar to the imprinting procedure employed for the MRI experiment, chicks were individually caged on the day of hatching with the imprinting object until day 3. The Red imprinted group was exposed for two days to the red flickering light ($N = 12$), while the Blue imprinted group to the blue flickering light ($N = 12$).

On day 3, all chicks were individually exposed to the pseudo-random sequence of red and blue colours that were employed for the stimulation inside the scanner (for details see section Acquisition and Pre-processing of fMRI data—Experimental task). Each chick saw 24 times its imprinting colour (red/blue) and 24 times the control colour (red/blue).

After the exposure, each chick was tested individually inside a running wheel to evaluate its colour preference. The test in the running wheel lasted a total of 10 min. Each colour was presented for 5 min. The sequence of colour presentation was counterbalanced between subjects.

The dependent variable measured was the distance (cm) covered by each subject toward the red and the blue. To estimate chicks' colour preferences, we calculated an index using the formula:

$$Colour\ preference = \frac{cm\ toward\ red}{cm\ toward\ red + cm\ toward\ blue}$$

This index could range between 0 (absolute preference for the blue) and 1 (absolute preference for the red), whereas 0.5 represented the absence of preference between the two colours.

## Statistics and reproducibility

To perform group inference in fMRI experiments, subject-level parameter estimates were taken into the second-level analysis using the mixed-effect model (FLAME1 + 2) to produce group-level estimates of each condition. FLAME 1 + 2 cluster-based approach has been used to threshold the group-level statistical maps for contrasts of interest with a cluster-defining voxel threshold of $p < 0.001$ (Z > 3.1) for red group and $p < 0.01$ (Z > 2.3) for blue group and entire timeseries analysis and Family Error Wise (FEW) cluster significance threshold of $p = 0.05$.

To estimate differences between the two imprinting groups in the colour preference experiment, we employed a two-tailed independent samples $t$-test. To estimate colour preference, we employed one-sample two-tailed $t$-test against chance (0.5).

## Reporting summary

Further information on research design is available in the Nature Portfolio Reporting Summary linked to this article.

## Data availability

All fMRI data for the chick imprinting and resting-state fMRI are available at (https://data.mendeley.com/datasets/w6cwvmbxwr/1)[100]. All data needed to evaluate the conclusions in the paper are present in the paper and/or the Supplementary Materials. The source data underlying Supplementary Fig. 2 is provided in Supplementary Data 1.

## Code availability

FSL software (https://fsl.fmrib.ox.ac.uk/fsl/fslwiki/, version 6.0.4) and MATLAB (2020b, MathWorks, USA) were used to process fMRI and behavioural data, respectively. Related processing codes can be found at https://github.com/mehdibehroozi/Imprinting-fMRI.

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

## Acknowledgements

This work was supported by the MIUR-DAAD Joint Mobility Program (project number 33538), the Deutsche Forschungsgemeinschaft (DFG, German Research Foundation) through grant SFB 874 (A1) project number 122679504 and SFB 1280 (A01, A08, and F02) project number 316803389. O.G. was in addition funded by DFG through Gu 227/16-1 and the European Research Council (ERC) under the European Union's Horizon 2020 research and innovation programme (ERC-2020-ADG, grant agreement No. [101021354, AVIAN MIND]). G.V. acknowledges grants from the European Research Council under the European Union's Seventh Framework Pro-gramme (FP7/2007–2013) Grant ERC-2011-ADG_20110406, Project no: 461 295517, PREMESOR), by Fondazione Caritro Grant Bio-marker DSA [40102839], and PRIN 2015. A.G. acknowledges funding by the European Research Council (ERC, DISCONN; no. 802371), the Brain and Behaviour Foundation (NARSAD Independent Investigator Grant #25861), the NIH (1R21MH116473-01A1) and the Telethon Foundation (GGP19177).

## Author contributions

Conceptualization: Onur Güntürkün and Giorgio Vallortigara. Experiment design: Mehdi Behroozi, Elena Lorenzi, Onur Güntürkün, and Giorgio Vallortigara. Data Collection: Mehdi Behroozi, Elena Lorenzi, and Sepideh Tabrik Methodology: Mehdi Behroozi and Sepideh Tabrik Resources: Martin

Tegenthoff, Alessandro Gozzi, Onur Güntürkün, and Giorgio Vallortigara Writing: All authors Visualization: Mehdi Behroozi, Elena Lorenzi, Onur Güntürkün, and Sepideh Tabrik.

## Funding

## Competing interests
The authors declare no competing interests.
