## [Transparent Peer Review file · Communications Biology]

Functional MRI of imprinting memory

Corresponding Author: Dr Mehdi Behroozi

Version 0:

Reviewer comments:

Reviewer #1

(Remarks to the Author)

The current study aimed at assessing brain activation, indexed by BOLD signals, for imprinting memory in awake newly hatched chicks. The new fMRI protocol is amazing and would open a new venue for developmental neuroscience and ethology.

However, the current study did not fully achieve the stated aims, primarily because the experimental design conflated imprinting memory acquisition and predisposition. With the wisdom of hindsight, the experiment could have adopted a pair of stimuli which are equally potent in inducing imprinting memory, which would dissociate neural responses based on predisposition, and those based on memory acquisition and retrieval. I understand that authors did a series of creative follow-up analyses to explore the brain regions which are likely to relate to each stage of imprinting memory formation, which provide certain level of unique and original contribution to the knowledge.

Otherwise, the experiment and analyses are conducted in high standard. Paper is written in accessible language, with clear expression and coherent structure.

Reviewer #2

(Remarks to the Author)

Functional MRI of imprinting memory: a new avenue for neurobiology of early learning

This article will be suitable for publication in Communications Biology after revisions regarding the interpretations of data.

Behroozi et al. present the data as the first example of fMRI study of the brain in awake newborn chicks. They used newly hatched chicks to identify the long-term storage of imprinting memories. They exposed chicks on either a preferred (red) or a non-preferred (blue) color object for two days to imprint them, and the chicks were tested with a sequence alternating the two colors in the scanner for fMRI study.

They established an entirely non-invasive awake fMRI protocol for the brains of newborn chicks and verified its utility.

They found that chicks imprinted on red color showed activities toward red color during the whole scanning in various brain regions, including the hippocampal formation, the medial striatum, the arcopallium, and the prefrontal-like nidopallium caudolaterale. They interpreted the activities as the responses reflecting imprinting memory retrieval.

When they used chicks imprinted on blue color, no significant activation signals were detected in the whole scanning. However, when they focused on the last 20 minutes of scanning, significant activities were detected toward the control color of red over the imprinting color blue. They interpreted the activities as the responses reflecting the acquisition phenomena of second imprinting memory.

The establishment of an entirely non-invasive awake fMRI protocol for the brains of newborn chicks is of considerable interest. The study's scientific impacts will be significant to researchers in the field of learning and memory.

Main point

The authors' statements and interpretations about the establishment of the non-invasive awake fMRI protocol and the scanning data using chicks imprinted on red color contain sufficient qualities for a publication.

I do not agree with the authors' interpretation of the data focusing on the last 20 minutes of scanning when chicks were imprinted on blue color. They interpreted the activities elicited by the red color during the scanning as responses reflecting the second imprinting memory acquisition. However, other interpretations are possible, and they should be mentioned in the manuscript.

In this experiment, there was no evidence of its involvement in acquiring first imprinting. The authors exposed chicks to a non-preferred (blue) color object for two days, but they did not detect significant activation signals against the imprinting color blue in the whole scanning, which means there were no indications of activities as traces of the first imprinting memory. Thus, it is not appropriate to term the activities against the control color red in the last 20 minutes of scanning the phenomena of the start or acquisition of the second imprinting.

The simple interpretation is that the imprinting using blue color was not successful for unexpected reasons or weakly imprinted in detecting the recall signals in the scanning of the fMRI study. The authors should mention a possibility in the discussion section that it may be the activity at the acquisition state of first imprinting on red color.

Reviewer #3

(Remarks to the Author)

The manuscript titled "Functional MRI of imprinting memory: a new avenue for the neurobiology of early learning" presents a method for identifying alterations in the neural network of chicks during filial imprinting. The manuscript, however, does not offer new evidence to shed light on the pathways involved in imprinting.

1. The authors did not provide enough details about the protocol they used to test chicken's preference for red and blue colours.
2. The manuscript describes the MRI protocol and recording method for a filial imprinting experiment in chicks. However, it failed to provide sufficient evidence to support the existing theories.
3. The discussed results seem to be rather simplistic. The authors did not observe any significant differences between imprinted and nonimprinted (Figure 2A). This suggests a possibility of either a failed experiment or an incorrect protocol being followed for imprinting. Imprinted memories are often subtle and only a small set of neurons are modified in the brain. Therefore, MRI may not be able to identify changes in such a low subset of neurons.
4. The manuscript states that the authors found no significant differences between the red and blue imported groups in regard to the red stimulus (line 176). However, the "idea" to support this failed experimental protocol is incorrect. The argument that "the blue-imprinted chicks exposed to the preferred colour red immediately started a process of secondary imprinting towards red colour" has no evidence and is not in line with the established protocol. All imprinting experiments test the effect of memory alterations in adults after conditioning at the early stages of development. The protocol followed in the manuscript appears to have carried out imprinting and testing on the chicks themselves.
5. The manuscript does not provide evidence supporting the claim that MRI-based techniques can identify connectome variations associated with memory.
6. In my opinion, the way the manuscript is to confuse the readers. As a general rule, scientific papers should clearly and concisely communicate the facts they are presenting, without requiring the reader to spend an excessive amount of time deciphering what is being said.

Version 1:

Reviewer comments:

Reviewer #2

(Remarks to the Author)

The revised manuscript is an important contribution to the society for the behavioral neuroscience and I recommend that it is accepted for publication.

Reviewer #3

(Remarks to the Author)

I still maintain my reservations about the study.

1. The provided protocol is very sketchy and does not specify whether it follows any established imprinting protocol for red and blue light. There is an established protocol for red light imprinting by Bradley et al (1981). According to this protocol, chicks were hatched and reared in darkness to approximately equal to 21 h when they were exposed to overhead illumination for 0.5 h and then to an imprinting stimulus (a pulsing red light) for 20 min. The chicks were then matched in pairs on the basis of their activity. One member of each pair was returned to the dark and the other was trained for a further

120 min.

2. In the reply, the authors just reiterated the protocol mentioned in the earlier version without giving any additional details or references to support it as a well-established protocol.

3. When the authors compared their behavioral assay to measure imprinting between red and blue, they did not obtain any statistical significance (see Figure 3). Even the imprinting data under red light shows huge variations, with some values falling below 0.5. This suggests that the imprinting protocol they followed is not be robust enough.

If the imprinting protocol itself is questionable, then all other data is irrelevant and not supportive of the working hypothesis.

Ref.

Bradley P, Horn G, Bateson P. Imprinting. An electron microscopic study of chick hyperstriatum ventrale. *Exp Brain Res.* 1981;41(2):115-20. doi: 10.1007/BF00236600. PMID: 7202608.

Version 2:

Reviewer comments:

Reviewer #3

(Remarks to the Author)

The manuscript by Behroozi et al. attempts to map the brain regions involved in filial imprinted memory using functional MRI. The approach is interesting but has serious flaws in their experimental approach.

1. I have mentioned in my first two reviews that the authors did not properly follow the imprinting protocol. Even though the authors have included the reference to Kovach 2008 in the methodology, the experiments conducted in the manuscript violate this protocol.

The fundamental rules of imprinting are as follows: one should choose a neutral conditioning stimulus for imprinting. After imprinting, the chick should show significant changes in behavior towards the stimulus compared to non-imprinted age-matched controls. Kovach and other researchers conducting experiments on imprinting use this standard approach. (Maekawa, F. et al. *BMC Neurosci.* 7, 75 (2006); Wang, S., Vasas, V., Freeland, L., Osorio, D. & Versace, E. *iScience* 27, 110195 (2024)).

Behroozi et al. used blue light as a control for imprinted red chicks, and red light as a control for imprinted blue chicks. On line 149, authors state, "For the preferred color group, the imprinting color (Imp) was red and the control (Cont) was blue, while for the non-preferred color group, the imprinting color was blue and the control was red."

This experimental approach is flawed as it does not include the results of a non-imprinted control, which is crucial for highly sensitive experiments like imprinting. It is unclear why the authors used non-imprinted chicks as the control.

2. The authors explained the results of the behavioral experiment in the following way: "These results confirmed the presence of no significant differences between the Red and the Blue imprinted groups with regard to the preference for the red stimulus." However, these results contradict the previously published findings (ref: Wang et al., *iScience* 2024), and the authors fail to provide justification. Instead, the authors are proposing a peculiar hypothesis. "These results might support the idea that Blue imprinted chicks exposed to the preferred colour red immediately started a process of secondary imprinting toward it inside the scanner." No scientific basis supports this argument. The authors heavily rely on hypothetical arguments in this manuscript without showing proper controls to justify their results.

3. In many instances, authors are using the > and < symbols throughout the manuscript. It is challenging to decipher the meaning of these symbols.

For example

Imp > Con line 151

RedImp + BlueImp > BlueCont + RedCont line 159

RedImp > BlueImp and BlueCont < RedCont contrasts line 166

We value the feedback provided by the Editor and Reviewers and recognize the significance of addressing their concerns. In response, we have implemented several amendments to the manuscript. A detailed point-by-point response letter follows.

Reviewer #1 (Remarks to the Author):

The current study aimed at assessing brain activation, indexed by BOLD signals, for imprinting memory in awake newly hatched chicks. The new fMRI protocol is amazing and would open a new venue for developmental neuroscience and ethology.

Thanks for your comment.

However, the current study did not fully achieve the stated aims, primarily because the experimental design conflated imprinting memory acquisition and predisposition. With the wisdom of hindsight, the experiment could have adopted a pair of stimuli which are equally potent in inducing imprinting memory, which would dissociate neural responses based on predisposition, and those based on memory acquisition and retrieval. I understand that authors did a series of creative follow-up analyses to explore the brain regions which are likely to relate to each stage of imprinting memory formation, which provide certain level of unique and original contribution to the knowledge.

We thank the Reviewer for the comment. We totally agree that the experimental design would have benefitted from using two equally preferred colours, helping this way disentangling the difference between “pure” imprinting related processes and “pure” predisposition mechanisms. However, the only colours that are equally preferred for imprinting in chicks are yellow, orange and red. Given the extreme similarity between these colours, we were however forced to use blue and red, two clearly distinguishable colours for we wanted to be sure that, during the test

phase in the scanner, chicks could clearly distinguish between the imprinting and the non-imprinting colour.

Otherwise, the experiment and analyses are conducted in high standard. Paper is written in accessible language, with clear expression and coherent structure.

We thank again the Reviewer for his comments.

Reviewer #2 (Remarks to the Author):

Functional MRI of imprinting memory: a new avenue for neurobiology of early learning
This article will be suitable for publication in Communications Biology after revisions regarding the interpretations of data.

Behroozi et al. present the data as the first example of fMRI study of the brain in awake newborn chicks. They used newly hatched chicks to identify the long-term storage of imprinting memories. They exposed chicks on either a preferred (red) or a non-preferred (blue) color object for two days to imprint them, and the chicks were tested with a sequence alternating the two colors in the scanner for fMRI study.

They established an entirely non-invasive awake fMRI protocol for the brains of newborn chicks and verified its utility.

They found that chicks imprinted on red color showed activities toward red color during the whole scanning in various brain regions, including the hippocampal formation, the medial striatum, the arcopallium, and the prefrontal-like nidopallium caudolaterale. They interpreted the activities as the responses reflecting imprinting memory retrieval.

When they used chicks imprinted on blue color, no significant activation signals were detected in the whole scanning. However, when they focused on the last 20 minutes of

scanning, significant activities were detected toward the control color of red over the imprinting color blue. They interpreted the activities as the responses reflecting the acquisition phenomena of second imprinting memory.

The establishment of an entirely non-invasive awake fMRI protocol for the brains of newborn chicks is of considerable interest. The study's scientific impacts will be significant to researchers in the field of learning and memory.

Main point

The authors' statements and interpretations about the establishment of the non-invasive awake fMRI protocol and the scanning data using chicks imprinted on red color contain sufficient qualities for a publication.

I do not agree with the authors' interpretation of the data focusing on the last 20 minutes of scanning when chicks were imprinted on blue color. They interpreted the activities elicited by the red color during the scanning as responses reflecting the second imprinting memory acquisition. However, other interpretations are possible, and they should be mentioned in the manuscript.

In this experiment, there was no evidence of its involvement in acquiring first imprinting. The authors exposed chicks to a non-preferred (blue) color object for two days, but they did not detect significant activation signals against the imprinting color blue in the whole scanning, which means there were no indications of activities as traces of the first imprinting memory.

Thus, it is not appropriate to term the activities against the control color red in the last 20 minutes of scanning the phenomena of the start or acquisition of the second imprinting.

The simple interpretation is that the imprinting using blue color was not successful for unexpected reasons or weakly imprinted in detecting the recall signals in the scanning of the

fMRI study. The authors should mention a possibility in the discussion section that it may be the activity at the acquisition state of first imprinting on red color.

We thank the Reviewer for the useful comment. We apologize for not being clear. To provide further evidence concerning the state of imprinting in the blue-imprinted group, we did an extra analysis for the first 10 min of scanning. In fact, these data seem to suggest that similar brain regions are active during the first minutes of scanning for blue imprinted chicks presented with blue and for red imprinted chicks presented with red. However, blue imprinted chicks show weaker activity in such brain regions when compared with red imprinted ones. From this evidence, we interpret our findings during the last 20 minutes of scanning in the blue imprinted group as the result of the initiation of a secondary imprinting phenomenon. Please see Figure R1, referring to the first 10 minutes of blue imprinted chicks presented with blue. As you can observe in the Figure, during the first 10 minutes of scanning, chicks imprinted with blue when presented with blue showed activity in the intermediate medial mesopallium (IMM), the hippocampus (Hp), the medial Striatum (MSt), the nidopallium caudolaterale (NCL) and the nucleus Taeniae of the Amygdala (TnA), similarly to what happens in red imprinted chicks when presented with red, albeit less strongly. This information has been added to the Results and Discussion and a figure has been added to Supplementary Materials (Figure R1). Nevertheless, given that we did not test for imprinting learning behaviorally before scanning (to not interfere with the data obtained in the MRI), we cannot be sure, as the Reviewer pointed out, that the phenomenon happening during the last 20 minutes of scanning is for sure the onset of the secondary imprinting, it could be as well the onset of a primary imprinting for red. We amended the discussion adding this possibility.

We modified the Results and Discussion as follows:

“As shown in Figures 2, 3, S3, and S4, this is due to chicks’ preference for red over blue (as demonstrated through the behavioural experiment), therefore we used the Imp > Cont contrast (blue > red colour) during the first 10 min and Cont > Imp contrast (red > blue colour) during the last 20 minutes of scanning, to investigate the memory retravel and memory formation phase of a new imprinting process²⁴ elicited by the presence of the preferred colour red. The results of the first 10 min scans indicate weak activity in the intermediate medial mesopallium (IMM), the hippocampus (Hp), the medial Striatum (MSt), the nidopallium caudolaterale (NCL) and the nucleus Taeniae of the Amygdala (TnA) when chicks imprinted on blue colour were initially presented with blue ($n = 8$, $Z = 1.9$ and $p < 0.05$ FEW corrected at the cluster level). However, as illustrated in Figure 4, the voxel-based group analysis during the last 20 min scans showed robust BOLD responses in different visual prosencephalic regions: the nucleus geniculatus lateralis pars dorsalis (GLd, which receives direct input from the retina²⁵), the right intermediate hyperpallium apicale (IHA, which primarily receives visual thalamic input²⁶), the right hyperpallium intercalatum (HI) and right hyperpallium densocellulare (HD), and bilaterally the hyperpallium apicale (HA, together with HD associative hubs of the thalamofugal pathway^{26,27}) of the thalamofugal pathway, bilaterally the nucleus rotundus (Rot, which is the primary thalamic input region of the tectofugal pathway). Also, parts of the auditory system were activated: bilaterally the ventromedial part of the Field-L complex and the right nucleus ovoidalis (OV), a thalamic auditory nucleus receiving direct input from the avian homologue of the inferior colliculus (torus semicircularis²⁸) that projects to Field-L. We detected significant activation clusters in the associative pallial regions nidopallium medial pars medialis (NMm) and bilaterally in the caudal intermediate medial mesopallium (IMM). Within the two interconnected Social Behavior Network and Mesolimbic Reward System, we detected a significant BOLD increase rightward in the bed nucleus of the stria terminalis (BNST), the nucleus accumbens (Ac) and the medial striatum (MSt), bilaterally

in the septum and leftward in the posterior pallial amygdala (PoA) and the ventromedial part of hippocampus (Hp-VM)."

Discussion: "Here we established a new non-invasive fMRI protocol to study awake brain activity in newly hatched domestic chicks in order to discover the neural pathways of imprinting and the identity of S'. After two days of imprinting training, with either a preferred (red) or a non-preferred (blue) colour, chicks were exposed to a sequence of the two stimulus colours inside the scanner. In Red imprinted chicks we found a network of brain regions probably associated with the long-term encoding and retrieval of imprinting memory. Interestingly, Blue imprinted chicks did not show such strong brain activity in these brain regions. To further explore the difference between the two groups, we conducted separate analyses for the initial 10 minutes and the final 20 minutes of the scanning for the Blue imprinted chicks. The analysis of the first 10 min unveiled that blue imprinted chicks when presented with blue did show an activation (albeit weaker) of the very same brain regions observed when red imprinted chicks were presented with red (Figure S3). Interestingly, during the last 20 minutes of scanning blue imprinted chicks showed a progressively increasing activity when presented with red in brain regions that we know from previous literature are associated with the very first phases of imprinting learning^{10-12,15,17,18,39,40}. We interpret these findings as follows: when the red colour is available, a new imprinting process begins toward it, as red is highly preferred by chicks (Figure 4). Such a phenomenon could be (i) a secondary imprinting process starting or (ii) the start of the first imprinting on red, given that the initial imprinting with blue was notably weaker or absent. In the following sections, we will refer to this as the acquisition phase of new imprinting."

Figure R1- BOLD response pattern during imprinting memory retrieval of blue imprinted colour. The high-resolution coronal slices at the different levels of an *ex-vivo* chick brain are in greyscale, while the contrast map represents the activation pattern during the presentation of the non-preferred imprinting object after imprinting learning has already occurred (Blue group, the contrast of blue light versus red light conditions during first 10 min of experiment, $n = 8$, $Z = 1.9$ and $p < 0.05$ FEW corrected at the cluster level). The results indicate weak activity in the intermediate medial mesopallium (IMM), the hippocampus (Hp), the medial Striatum (MSt), the nidopallium caudolaterale (NCL) and the nucleus Taeniae of the Amygdala (TnA) when chicks imprinted on blue colour were initially presented to blue. The results are, similar to what happens in red imprinted chicks when presented with red colour (Figure 5). The corresponding abbreviations of ROIs are listed in Table S1.

Reviewer #3 (Remarks to the Author):

The manuscript titled “Functional MRI of imprinting memory: a new avenue for the neurobiology of early learning” presents a method for identifying alterations in the neural network of chicks during filial imprinting. The manuscript, however, does not offer new evidence to shed light on the pathways involved in imprinting.

1. The authors did not provide enough details about the protocol they used to test chicken’s preference for red and blue colours.

We tried to specify as clearly as possible these details, as shown below:

Chicks were individually caged on the day of hatching with the imprinting object until day 3. The Red imprinted group was exposed for two days to the red flickering light (N =12), while the Blue imprinted group to the blue flickering light (N = 12). [All the details are reported in the Imprinting section of the Materials and Methods.]

On day 3, all chicks were individually exposed to a pseudo random sequence of red and blue colours inside the scanner (for details see section Acquisition and Pre-processing of fMRI data - Experimental task). Each chick saw 24 times its imprinting colour (red/blue) and 24 times the control colour (red/blue).

After the exposure, each chick was tested individually inside a running wheel to evaluate its colour preference. The test in the running wheel lasted a total of 10 minutes. Each colour was presented for 5 minutes. The sequence of colour presentation was counterbalanced between subjects.

The dependent variable measured was the distance (cm) covered by each subject toward the red and the blue. To estimate chicks' colour preference, we calculated an index using the formula:

$$\text{Colour preference} = \frac{\text{cm toward red}}{\text{cm toward red} + \text{cm toward blue}}$$

This index could range between 0 (absolute preference for the blue) and 1 (absolute preference for the red), whereas 0.5 represented the absence of preference between the two colours.

To estimate differences between the two imprinting groups we employed a two-tailed independent samples *t*-test. To estimate colour preference, we employed one-sample two-tailed *t*-test against chance (0.5).

2. The manuscript describes the MRI protocol and recording method for a filial imprinting experiment in chicks. However, it failed to provide sufficient evidence to support the existing theories.

We would be really grateful to the Reviewer if she/he could mention the evidence he believes is needed and the existing theories she/he believes we missed to support. We will be happy to answer specifically to the Reviewer and update the manuscript accordingly.

3. The discussed results seem to be rather simplistic. The authors did not observe any significant differences between imprinted and nonimprinted (Figure 2A). This suggests a possibility of either a failed experiment or an incorrect protocol being followed for imprinting. Imprinted memories are often subtle and only a small set of neurons are modified in the brain. Therefore, MRI may not be able to identify changes in such a low subset of neurons.

We apologize for any confusion that may have arisen. We initially hypothesized that the contrast ($\text{Red}_{\text{Imp}} + \text{Blue}_{\text{Imp}} > \text{Blue}_{\text{Cont}} + \text{Red}_{\text{Cont}}$) would reveal networks associated with memory retrieval. This contrast combined the imprinting colours (red for red imprinted chicks and blue presentation for blue imprinted chicks) and was subtracted from the control colours (blue for red imprinted chicks and red for blue imprinted chicks), to illustrate the general effect of imprinting. However, this contrast did not yield any significantly activated cluster in the chick brain. Upon closer examination, we realized that the activation patterns for both contrasts, $\text{Red}_{\text{Imp}} + \text{Blue}_{\text{Imp}} > \text{baseline}$ and $\text{Blue}_{\text{Cont}} + \text{Red}_{\text{Cont}} > \text{baseline}$ were remarkably similar (refer to Figure 2A). This suggested to us that the introduction of the highly preferred red color in the Blue group acted as an imprinting stimulus. The brain's response to the red color in the Blue group closely resembled that of the Red group, indicating the onset of new imprinting. To discern whether this was secondary imprinting or a failure of blue imprinting, we conducted separate analyses for the first 10 (please see the response for Reviewer #2 for more details) and the last 20 minutes. The results revealed a blue imprinting effect in the Blue group during the

first 10 min (Figure R1, see also the response to comments #1 of Editor for more details), with a new imprinting occurring after the introduction of the red color during the test session.

In summary, significant differences were observed between imprinted and non-imprinted colors (see Figure 4 for the last 20 minutes of the Blue group, Figure 5 for the Red group, and Figure S3 for the first 10 minutes of the Blue group). Notably, our behavioral data align with fMRI findings: the behavioral experiment revealed a strong preference for red in Red imprinted chicks but no preference for blue in Blue imprinted chicks.

Although, brain changes related to imprinting memory could be subtle, in the present experiment we were able to find significant changes for imprinting memory retrieval and long-term storage in the red imprinted group. We also managed to find significant brain changes related to the onset of imprinting learning in the blue group exposed to red (please see the response for Reviewer #2 for more details).

4. The manuscript states that the authors found no significant differences between the red and blue imported groups in regard to the red stimulus (line 176). However, the “idea” to support this failed experimental protocol is incorrect. The argument that “the blue-imprinted chicks exposed to the preferred colour red immediately started a process of secondary imprinting towards red colour” has no evidence and is not in line with the established protocol. All imprinting experiments test the effect of memory alterations in adults after conditioning at the early stages of development. The protocol followed in the manuscript appears to have carried out imprinting and testing on the chicks themselves. We reiterate (see responses to Reviewer #2) that we were able to document also imprinting on blue color in the initial 10 min. Note also that filial imprinting experiments in chicks (*Gallus gallus*) usually test memory changes triggered during the first days of life not adults (please

see just some examples here (Bolhuis (1991), Regolin and Vallortigara (1995), Vallortigara et al., (1998), Lemaire et al. (2021), Miura and Matsushima (2016), Bolhuis (1999)). For more details please refer to response to Reviewer #2.

References

Bolhuis, J. J. (1991). Mechanisms of avian imprinting: a review. *Biological Reviews*, 66(4), 303-345.

Regolin, L., & Vallortigara, G. (1995). Perception of partly occluded objects by young chicks. *Perception & psychophysics*, 57, 971-976.

Vallortigara, G., Regolin, L., Rigoni, M., & Zanforlin, M. (1998). Delayed search for a concealed imprinted object in the domestic chick. *Animal Cognition*, 1, 17-24.

Lemaire, B. S., Rugani, R., Regolin, L., & Vallortigara, G. (2021). Response of male and female domestic chicks to change in the number (quantity) of imprinting objects. *Learning & Behavior*, 49, 54-66.

Miura, M., & Matsushima, T. (2016). Biological motion facilitates filial imprinting. *Animal Behaviour*, 116, 171-180.

Bolhuis, J. J. (1999). Early learning and the development of filial preferences in the chick. *Behavioural brain research*, 98(2), 245-252.

5. The manuscript does not provide evidence supporting the claim that MRI-based techniques can identify connectome variations associated with memory.

We adopted the Discussion as follows to provide evidence supporting fMRI application in identifying connectome variations associated with different forms of memory:

“fMRI has been used in several studies to investigate the connection between connectome variations and memory. A recent study in songbirds highlights fMRI's role in tracking song memory development in zebra finch's post-tutorial sessions³⁵. The study revealed permanent neural activity changes in auditory perception and song learning, highlighting early sensory

experiences. Gazzaley and Nobre³⁶ explored the neural basis of working memory encoding and maintenance using fMRI. Rahm et al.³⁷ characterized the neural basis of visual working memory recognition using fMRI by varying recognition needs and similarity between probe items and memory contents. Yang et al.³⁸ proposed an enhanced connectome-based predictive modeling approach, which showed strong applicability across different cognitive processes and could predict working memory performance in healthy individuals. These findings underscore the potential of fMRI in understanding brain processes that underpin cognitive abilities.”

6. In my opinion, the way the manuscript is to confuse the readers. As a general rule, scientific papers should clearly and concisely communicate the facts they are presenting, without requiring the reader to spend an excessive amount of time deciphering what is being said.

We are sorry about that. We did our best to make the manuscript as clear as possible in the revised version.

Reviewers' comments:

Reviewer #2 (Remarks to the Author):

The revised manuscript is an important contribution to the society for the behavioral neuroscience and I recommend that it is accepted for publication.

Thank you very much for your positive and kind comments!

Reviewer #3 (Remarks to the Author):

I still maintain my reservations about the study.

1. The provided protocol is very sketchy and does not specify whether it follows any established imprinting protocol for red and blue light. There is an established protocol for red light imprinting by Bradley et al (1981). According to this protocol, chicks were hatched and reared in darkness to approximately equal to 21 h when they were exposed to overhead illumination for 0.5 h and then to an imprinting stimulus (a pulsing red light) for 20 min. The chicks were then matched in pairs on the basis of their activity. One member of each pair was returned to the dark and the other was trained for a further 120 min.

2. In the reply, the authors just reiterated the protocol mentioned in the earlier version without giving any additional details or references to support it as a well-established protocol.

We are sorry that the Reviewer found the protocol not very well described. In the revised version of the paper we tried to provide all details and of course we are ready to add anything the Reviewer thinks is necessary and that we may have not considered (see pages 43 and 44 of Materials and Methods). Our lab worked using imprinting procedures for somewhat the last 35 years and we published widely on this topic (actually, we believe that after the passing away of Professor Gabriel Horn we have been the main lab studying imprinting on the behavioural and neurobiological side in recent years, see e.g. our recent monography published by MIT Press: *Born Knowing: Imprinting and the Origins of Knowledge*, 2021, cited in the paper). Thus, we know of course Bradley et al but also other established protocols. More in details, Kovach (1971) first showed that prolonged exposure (during the first two days after hatch) of different coloured lights is capable of eliciting stable filial imprinting in chicks. Here we used a similar paradigm. During the first two days of life chicks were individually exposed to either red or blue light for 12 hours per day for a total exposure time of 24 hours per each subject. Many studies investigating filial imprinting found that even shorter exposure times can be actually effective to elicit imprinting (see e.g. for work carried out in our lab: Lemaire et al. (2021). Stability and individual variability of social attachment in imprinting. *Scientific Reports* 11: 7914.). However, here we used prolonged exposure in order to be sure that imprinting was learned and stored in the long-term memory. We added the reference to Kovach's study in the revised manuscript.

Joseph K. Kovach. (1971). Effectiveness of Different Colors in the Elicitation and Development of Approach Behavior in Chicks. *Behaviour*, 38(1-2), 154–168. doi:10.2307/4533367

3. When the authors compared their behavioral assay to measure imprinting between red and blue, they did not obtain any statistical significance (see Figure 3). Even the imprinting data

under red light shows huge variations, with some values falling below 0.5. This suggests that the imprinting protocol they followed is not be robust enough.

We respectfully disagree on that. Nine out of 12 chicks imprinted on red showed a clear preference for red well above chance, 1 was at chance, and 2 preferred blue (exactly as Kovach 1991 already found). Six out of 12 chicks imprinted on blue showed a preference for red above chance, 1 was at chance, and 5 preferred blue. These data clearly support the fact that when tested in the scanner, red imprinted chicks maintained their imprinting preference and blue imprinted chicks started to lose it in favour of red. It seems to us that the misunderstanding here arises from the fact that the measure we reported is a *relative* preference for the two stimuli. We calculated a ratio of preference following this formula: (cm run toward red)/(cm run toward red + cm run toward blue). Values range from a maximum of 1 (chick that ran only for the red) and 0 (chick that ran only for the blue). This way of computing chicks' behaviour does not take into account the fact that all chicks (apart from only 2 – one imprinted with red and one imprinted with blue) ran toward their imprinting stimulus when presented with it (for the original raw data please see the table provided below). Thus, again, imprinting does occur for both colours, as showed by the approach response, but chicks have (as expected) a relative preference for red colour.

Subject id	Imprinting colour	First colour presented at test	Distance run toward Red (cm)	Distance run toward Blue (cm)	Ratio preference (approximated)
1	Red	Red	2923	882	0.8
2	Red	Blue	0	0	0.5
3	Red	Red	412	0	1
4	Red	Blue	134	336	0.3
5	Blue	Red	1403	748	0.7
6	Blue	Blue	286	269	0.5
7	Blue	Red	210	34	0.9
8	Blue	Blue	252	470	0.4
9	Red	Red	1714	445	0.8
10	Red	Blue	3553	118	1
11	Red	Red	84	25	0.8
12	Red	Blue	50	0	1
13	Blue	Red	160	0	1
14	Blue	Blue	386	546	0.4
15	Blue	Red	2285	840	0.7
16	Blue	Blue	151	67	0.7
17	Red	Red	17	25	0.4
18	Red	Blue	387	319	0.6
19	Red	Red	1705	302	0.9
20	Red	Blue	588	218	0.7
21	Blue	Red	3902	5813	0.4
22	Blue	Blue	202	2486	0.1
23	Blue	Red	588	395	0.6
24	Blue	Blue	260	378	0.4

If the imprinting protocol itself is questionable, then all other data is irrelevant and not supportive of the working hypothesis.

We hope to have now better clarified that the imprinting protocol we used (fully described in the Material and Methods) does follow classical and well-established procedures. All the literature on imprinting converges on the presence of two phenomena: the learning process itself (as a result of the exposure to the stimulus) and the presence of innate preferences (for colour in this case, though of course innate preferences for other stimulus properties have been found starting from the seminal work by Horn and colleagues). Thus, there is a striking correspondence between the behavioural data and the data collected in the scanner.

Revision3:

(1) Better emphasize your core result that similar brain regions are active during the first 10 minutes of the test session for blue-imprinted chicks presented with blue (albeit with weaker intensity) and for red-imprinted chicks presented with red; and that during this period, imprinting does occur for both colours and that the behavioral data align with the fMRI findings.

We have better emphasized this as follows (p.5 lines 77-90): *“We exposed (imprinted) chicks on either a preferred (red) or a non-preferred (blue) colour. After exposure, awake chicks were tested with a sequence alternating the two colours in the scanner. We could demonstrate that chicks imprinted on red colour showed activity in pallial and subpallial brain regions involved with storage and memory retrieval, such as the medial striatum, the arcopallium, the hippocampus, and the nidopallium caudolaterale (a presumed avian equivalent of mammalian prefrontal cortex). Chicks imprinted on blue showed less activity in the same regions; however, during the last 20 minutes of scanning when presented with the red, these chicks showed activity in the mesopallium and in the Social Behavior Network. The first exposure to the colour red, a predisposed feature for social attachment, thus started a process of secondary imprinting, activating a brain network known to be involved in socially predisposed features at birth.”*

(2) Address Reviewer #3's comment 2 regarding the contradictions between the results and prior work by providing a more detailed explanation to strengthen your hypothesis.

Referee's comment: 2. The authors explained the results of the behavioral experiment in the following way: "These results confirmed the presence of no significant differences between the Red and the Blue imprinted groups with regard to the preference for the red stimulus. "However, these results contradict the previously published findings (ref: Wang et al., iScience 2024), and the authors fail to provide justification. Instead, the authors are proposing a peculiar hypothesis. "These results might support the idea that Blue imprinted chicks exposed to the preferred colour red immediately started a process of secondary imprinting toward it inside the scanner. "No scientific basis supports this argument. The authors heavily rely on hypothetical arguments in this manuscript without showing proper controls to justify their results.

We kindly disagree with the Reviewer. Wang et al., 2024 imprinted chicks on different colours and tested their preference between the imprinted colour and a different colour but on the same continuum of the imprinted colour (e.g. different shades between red and yellow). Thus, in the mentioned study, blue imprinted chicks were never tested for their preference for red nor vice versa, but rather red was compared with shades of red and blue with shades of blue. Please note also that the preference for red over blue in imprinting is not a hypothetical argument but it is a fact supported by a large literature, which is by the way also quoted in the Wang et al paper (see e.g. references Lemaire et al., 2021; Versace et al., 2017; Miura et al., 2020; Rosa-Salva et al., 2010). We have clarified this in the revised version and added references in support.

Lemaire, B. S., Rucco, D., Josserand, M., Vallortigara, G., & Versace, E. (2021). Stability and individual variability of social attachment in imprinting. *Scientific Reports*, 11(1), 7914.

Versace, E., Fracasso, I., Baldan, G., Dalle Zotte, A., & Vallortigara, G. (2017). Newborn chicks show inherited variability in early social predispositions for hen-like stimuli. *Scientific Reports*, 7(1), 40296.

Miura, M., Nishi, D., & Matsushima, T. (2020). Combined predisposed preferences for colour and biological motion make robust development of social attachment through imprinting. *Animal cognition*, 23, 169-188.

Rosa-Salva, O., Regolin, L., & Vallortigara, G. (2010). Faces are special for newly hatched chicks: evidence for inborn domain-specific mechanisms underlying spontaneous preferences for face-like stimuli. *Developmental science*, 13(4), 565-577.

(3) Address Reviewer #3's comment 3 by clearly defining the terms ">" and "<" upon their first use in the manuscript.

We have clarified the meaning of ">" upon its first use. Specifically, ">" indicates a greater or stronger activity for the first condition compared to the second, while "<" indicates less activity for the first condition compared to the second. We updated the sentences in lines 144 and 159 as follows:

“To identify the long-term storage of imprinting memory, we first used the contrast of Imp > Cont, where 'Imp > Cont' indicates greater activity for the implicit condition (Imp) compared to the control condition (Cont), by combining both groups in a conventional generalised linear model (GLM) based statistical analysis.”

“As illustrated in Figures 2B and 2C, robust BOLD activation patterns were found within the telencephalon for the contrasts: Red_{Imp} > Blue_{Imp} (indicating greater

activity in Red_{Imp} than Blue_{Imp}) and Blue_{Cont} < Red_{Cont} (indicating less activity in Blue_{Cont} than Red_{Cont}) contrasts.”

Revision2:

Reviewers' comments:

Reviewer #2 (Remarks to the Author):

The revised manuscript is an important contribution to the society for the behavioral neuroscience and I recommend that it is accepted for publication.

Thank you very much for your positive and kind comments!

Reviewer #3 (Remarks to the Author):

I still maintain my reservations about the study.

1. The provided protocol is very sketchy and does not specify whether it follows any established imprinting protocol for red and blue light. There is an established protocol for red light imprinting by Bradley et al (1981). According to this protocol, chicks were hatched and reared in darkness to approximately equal to 21 h when they were exposed to overhead illumination for 0.5 h and then to an imprinting stimulus (a pulsing red light) for 20 min. The chicks were then matched in pairs on the basis of their activity. One member of each pair was returned to the dark and the other was trained for a further 120 min.
2. In the reply, the authors just reiterated the protocol mentioned in the earlier

version without giving any additional details or references to support it as a well-established protocol.

We are sorry that the Reviewer found the protocol not very well described. In the revised version of the paper we tried to provide all details and of course we are ready to add anything the Reviewer thinks is necessary and that we may have not considered (see pages 43 and 44 of Materials and Methods). Our lab worked using imprinting procedures for somewhat the last 35 years and we published widely on this topic (actually, we believe that after the passing away of Professor Gabriel Horn we have been the main lab studying imprinting on the behavioural and neurobiological side in recent years, see e.g. our recent monography published by MIT Press: *Born Knowing: Imprinting and the Origins of Knowledge*, 2021, cited in the paper). Thus, we know of course Bradley et al but also other established protocols. More in details, Kovach (1971) first showed that prolonged exposure (during the first two days after hatch) of different coloured lights is capable of eliciting stable filial imprinting in chicks. Here we used a similar paradigm. During the first two days of life chicks were individually exposed to either red or blue light for 12 hours per day for a total exposure time of 24 hours per each subject. Many studies investigating filial imprinting found that even shorter exposure times can be actually effective to elicit imprinting (see e.g. for work carried out in our lab: Lemaire et al. (2021). Stability and individual variability of social attachment in imprinting. *Scientific Reports* 11: 7914.). However, here we used prolonged exposure in order to be sure that imprinting was learned and stored in the long-term memory. We added the reference to Kovach's study in the revised manuscript.

Joseph K. Kovach. (1971). Effectiveness of Different Colors in the Elicitation and Development of Approach Behavior in Chicks. *Behaviour*, 38(1-2), 154–168.
doi:10.2307/4533367

3. When the authors compared their behavioral assay to measure imprinting between red and blue, they did not obtain any statistical significance (see Figure 3). Even the imprinting data under red light shows huge variations, with some values falling below 0.5. This suggests that the imprinting protocol they followed is not be robust enough.

We respectfully disagree on that. Nine out of 12 chicks imprinted on red showed a clear preference for red well above chance, 1 was at chance, and 2 preferred blue (exactly as Kovach 1991 already found). Six out of 12 chicks imprinted on blue showed a preference for red above chance, 1 was at chance, and 5 preferred blue. These data clearly support the fact that when tested in the scanner, red imprinted chicks maintained their imprinting preference and blue imprinted chicks started to lose it in favour of red. It seems to us that the misunderstanding here arises from the fact that the measure we reported is a *relative* preference for the two stimuli. We calculated a ratio of preference following this formula: $(\text{cm run toward red}) / (\text{cm run toward red} + \text{cm run toward blue})$. Values range from a maximum of 1 (chick that ran only for the red) and 0 (chick that ran only for the blue). This way of computing chicks' behaviour does not take into account the fact that all chicks (apart from only 2 – one imprinted with red and one imprinted with blue) ran toward their imprinting stimulus when presented with it (for the original raw data please see the table provided below). Thus, again,

imprinting does occur for both colours, as showed by the approach response, but chicks have (as expected) a relative preference for red colour.

Subject id	Imprinting colour	First colour presented at test	Distance run toward Red (cm)	Distance run toward Blue (cm)	Ratio preference (approximated)
1	Red	Red	2923	882	0.8
2	Red	Blue	0	0	0.5
3	Red	Red	412	0	1
4	Red	Blue	134	336	0.3
5	Blue	Red	1403	748	0.7
6	Blue	Blue	286	269	0.5
7	Blue	Red	210	34	0.9
8	Blue	Blue	252	470	0.4
9	Red	Red	1714	445	0.8
10	Red	Blue	3553	118	1
11	Red	Red	84	25	0.8
12	Red	Blue	50	0	1
13	Blue	Red	160	0	1
14	Blue	Blue	386	546	0.4
15	Blue	Red	2285	840	0.7
16	Blue	Blue	151	67	0.7
17	Red	Red	17	25	0.4
18	Red	Blue	387	319	0.6

19	Red	Red	1705	302	0.9
20	Red	Blue	588	218	0.7
21	Blue	Red	3902	5813	0.4
22	Blue	Blue	202	2486	0.1
23	Blue	Red	588	395	0.6
24	Blue	Blue	260	378	0.4

If the imprinting protocol itself is questionable, then all other data is irrelevant and not supportive of the working hypothesis.

We hope to have now better clarified that the imprinting protocol we used (fully described in the Material and Methods) does follow classical and well-established procedures. All the literature on imprinting converges on the presence of two phenomena: the learning process itself (as a result of the exposure to the stimulus) and the presence of innate preferences (for colour in this case, though of course innate preferences for other stimulus properties have been found starting from the seminal work by Horn and colleagues). Thus, there is a striking correspondence between the behavioural data and the data collected in the scanner.

Revision1:

We value the feedback provided by the Editor and Reviewers and recognize the significance of addressing their concerns. In response, we have implemented several amendments to the manuscript. A detailed point-by-point response letter follows.

Reviewer #1 (Remarks to the Author):

The current study aimed at assessing brain activation, indexed by BOLD signals, for imprinting memory in awake newly hatched chicks. The new fMRI protocol is amazing and would open a new venue for developmental neuroscience and ethology.

Thanks for your comment.

However, the current study did not fully achieve the stated aims, primarily because the experimental design conflated imprinting memory acquisition and predisposition. With the wisdom of hindsight, the experiment could have adopted a pair of stimuli which are equally potent in inducing imprinting memory, which would dissociate neural responses based on predisposition, and those based on memory acquisition and retrieval. I understand that authors did a series of creative follow-up analyses to explore the brain regions which are likely to relate to each stage of imprinting memory formation, which provide certain level of unique and original contribution to the knowledge.

We thank the Reviewer for the comment. We totally agree that the experimental design would have benefitted from using two equally preferred colours, helping this way disentangling the difference between “pure” imprinting related processes and “pure” predisposition mechanisms. However, the only colours that are equally preferred for imprinting in chicks are yellow, orange and red. Given the extreme similarity between these colours, we were however forced to use

blue and red, two clearly distinguishable colours for we wanted to be sure that, during the test phase in the scanner, chicks could clearly distinguish between the imprinting and the non-imprinting colour.

Otherwise, the experiment and analyses are conducted in high standard. Paper is written in accessible language, with clear expression and coherent structure.

We thank again the Reviewer for his comments.

Reviewer #2 (Remarks to the Author):

Functional MRI of imprinting memory: a new avenue for neurobiology of early learning

This article will be suitable for publication in Communications Biology after revisions regarding the interpretations of data.

Behroozi et al. present the data as the first example of fMRI study of the brain in awake newborn chicks. They used newly hatched chicks to identify the long-term storage of imprinting memories. They exposed chicks on either a preferred (red) or a non-preferred (blue) color object for two days to imprint them, and the chicks were tested with a sequence alternating the two colors in the scanner for fMRI study.

They established an entirely non-invasive awake fMRI protocol for the brains of newborn chicks and verified its utility.

They found that chicks imprinted on red color showed activities toward red color during the whole scanning in various brain regions, including the hippocampal formation, the medial striatum, the arcopallium, and the prefrontal-like nidopallium caudolaterale. They interpreted the activities as the responses reflecting imprinting memory retrieval.

When they used chicks imprinted on blue color, no significant activation signals were

detected in the whole scanning. However, when they focused on the last 20 minutes of scanning, significant activities were detected toward the control color of red over the imprinting color blue. They interpreted the activities as the responses reflecting the acquisition phenomena of second imprinting memory.

The establishment of an entirely non-invasive awake fMRI protocol for the brains of newborn chicks is of considerable interest. The study's scientific impacts will be significant to researchers in the field of learning and memory.

Main point

The authors' statements and interpretations about the establishment of the non-invasive awake fMRI protocol and the scanning data using chicks imprinted on red color contain sufficient qualities for a publication.

I do not agree with the authors' interpretation of the data focusing on the last 20 minutes of scanning when chicks were imprinted on blue color. They interpreted the activities elicited by the red color during the scanning as responses reflecting the second imprinting memory acquisition. However, other interpretations are possible, and they should be mentioned in the manuscript.

In this experiment, there was no evidence of its involvement in acquiring first imprinting. The authors exposed chicks to a non-preferred (blue) color object for two days, but they did not detect significant activation signals against the imprinting color blue in the whole scanning, which means there were no indications of activities as traces of the first imprinting memory. Thus, it is not appropriate to term the activities against the control color red in the last 20 minutes of scanning the phenomena of the start or acquisition of the second imprinting. The simple interpretation is that the imprinting using blue color was not successful for unexpected reasons or weakly imprinted in detecting the recall signals in the scanning of the

fMRI study. The authors should mention a possibility in the discussion section that it may be the activity at the acquisition state of first imprinting on red color.

We thank the Reviewer for the useful comment. We apologize for not being clear. To provide further evidence concerning the state of imprinting in the blue-imprinted group, we did an extra analysis for the first 10 min of scanning. In fact, these data seem to suggest that similar brain regions are active during the first minutes of scanning for blue imprinted chicks presented with blue and for red imprinted chicks presented with red. However, blue imprinted chicks show weaker activity in such brain regions when compared with red imprinted ones. From this evidence, we interpret our findings during the last 20 minutes of scanning in the blue imprinted group as the result of the initiation of a secondary imprinting phenomenon. Please see Figure R1, referring to the first 10 minutes of blue imprinted chicks presented with blue. As you can observe in the Figure, during the first 10 minutes of scanning, chicks imprinted with blue when presented with blue showed activity in the intermediate medial mesopallium (IMM), the hippocampus (Hp), the medial Striatum (MSt), the nidopallium caudolaterale (NCL) and the nucleus Taeniae of the Amygdala (TnA), similarly to what happens in red imprinted chicks when presented with red, albeit less strongly. This information has been added to the Results and Discussion and a figure has been added to Supplementary Materials (Figure R1). Nevertheless, given that we did not test for imprinting learning behaviorally before scanning (to not interfere with the data obtained in the MRI), we cannot be sure, as the Reviewer pointed out, that the phenomenon happening during the last 20 minutes of scanning is for sure the onset of the secondary imprinting, it could be as well the onset of a primary imprinting for red. We amended the discussion adding this possibility.

We modified the Results and Discussion as follows:

“As shown in Figures 2, 3, S3, and S4, this is due to chicks’ preference for red over blue (as demonstrated through the behavioural experiment), therefore we used the Imp > Cont contrast (blue > red colour) during the first 10 min and Cont > Imp contrast (red > blue colour) during the last 20 minutes of scanning, to investigate the memory retravel and memory formation phase of a new imprinting process²⁴ elicited by the presence of the preferred colour red. The results of the first 10 min scans indicate weak activity in the intermediate medial mesopallium (IMM), the hippocampus (Hp), the medial Striatum (MSt), the nidopallium caudolaterale (NCL) and the nucleus Taeniae of the Amygdala (TnA) when chicks imprinted on blue colour were initially presented with blue (n = 8, Z = 1.9 and p < 0.05 FEW corrected at the cluster level). However, as illustrated in Figure 4, the voxel-based group analysis during the last 20 min scans showed robust BOLD responses in different visual prosencephalic regions: the nucleus geniculatus lateralis pars dorsalis (GLd, which receives direct input from the retina²⁵), the right intermediate hyperpallium apicale (IHA, which primarily receives visual thalamic input²⁶), the right hyperpallium intercalatum (HI) and right hyperpallium densocellulare (HD), and bilaterally the hyperpallium apicale (HA, together with HD associative hubs of the thalamofugal pathway^{26,27}) of the thalamofugal pathway, bilaterally the nucleus rotundus (Rot, which is the primary thalamic input region of the tectofugal pathway). Also, parts of the auditory system were activated: bilaterally the ventromedial part of the Field-L complex and the right nucleus ovoidalis (OV), a thalamic auditory nucleus receiving direct input from the avian homologue of the inferior colliculus (torus semicircularis²⁸) that projects to Field-L. We detected significant activation clusters in the associative pallial regions nidopallium medial pars medialis (NMm) and bilaterally in the caudal intermediate medial mesopallium (IMM). Within the two interconnected Social Behavior Network and Mesolimbic Reward System, we detected a significant BOLD increase rightward in the bed nucleus of the stria terminalis (BNST), the nucleus accumbens (Ac) and the medial striatum (MSt), bilaterally

in the septum and leftward in the posterior pallial amygdala (PoA) and the ventromedial part of hippocampus (Hp-VM)."

Discussion: "Here we established a new non-invasive fMRI protocol to study awake brain activity in newly hatched domestic chicks in order to discover the neural pathways of imprinting and the identity of S'. After two days of imprinting training, with either a preferred (red) or a non-preferred (blue) colour, chicks were exposed to a sequence of the two stimulus colours inside the scanner. In Red imprinted chicks we found a network of brain regions probably associated with the long-term encoding and retrieval of imprinting memory. Interestingly, Blue imprinted chicks did not show such strong brain activity in these brain regions. To further explore the difference between the two groups, we conducted separate analyses for the initial 10 minutes and the final 20 minutes of the scanning for the Blue imprinted chicks. The analysis of the first 10 min unveiled that blue imprinted chicks when presented with blue did show an activation (albeit weaker) of the very same brain regions observed when red imprinted chicks were presented with red (Figure S3). Interestingly, during the last 20 minutes of scanning blue imprinted chicks showed a progressively increasing activity when presented with red in brain regions that we know from previous literature are associated with the very first phases of imprinting learning^{10-12,15,17,18,39,40}. We interpret these findings as follows: when the red colour is available, a new imprinting process begins toward it, as red is highly preferred by chicks (Figure 4). Such a phenomenon could be (i) a secondary imprinting process starting or (ii) the start of the first imprinting on red, given that the initial imprinting with blue was notably weaker or absent. In the following sections, we will refer to this as the acquisition phase of new imprinting."

Figure R1- BOLD response pattern during imprinting memory retrieval of blue imprinted colour. The high-resolution coronal slices at the different levels of an *ex-vivo* chick brain are in greyscale, while the contrast map represents the activation pattern during the presentation of the non-preferred imprinting object after imprinting learning has already occurred (Blue group, the contrast of blue light versus red light conditions during first 10 min of experiment, $n = 8$, $Z = 1.9$ and $p < 0.05$ FEW corrected at the cluster level). The results indicate weak activity in the intermediate medial mesopallium (IMM), the hippocampus (Hp), the medial Striatum (MSt), the nidopallium caudolaterale (NCL) and the nucleus Taeniae of the Amygdala (TnA) when chicks imprinted on blue colour were initially presented to blue. The results are, similar to what happens in red imprinted chicks when presented with red colour (Figure 5). The corresponding abbreviations of ROIs are listed in Table S1.

Reviewer #3 (Remarks to the Author):

The manuscript titled “Functional MRI of imprinting memory: a new avenue for the neurobiology of early learning” presents a method for identifying alterations in the neural network of chicks during filial imprinting. The manuscript, however, does not offer new evidence to shed light on the pathways involved in imprinting.

1. The authors did not provide enough details about the protocol they used to test chicken’s preference for red and blue colours.

We tried to specify as clearly as possible these details, as shown below:

Chicks were individually caged on the day of hatching with the imprinting object until day 3. The Red imprinted group was exposed for two days to the red flickering light (N =12), while the Blue imprinted group to the blue flickering light (N = 12). [All the details are reported in the Imprinting section of the Materials and Methods.]

On day 3, all chicks were individually exposed to a pseudo random sequence of red and blue colours inside the scanner (for details see section Acquisition and Pre-processing of fMRI data - Experimental task). Each chick saw 24 times its imprinting colour (red/blue) and 24 times the control colour (red/blue).

After the exposure, each chick was tested individually inside a running wheel to evaluate its colour preference. The test in the running wheel lasted a total of 10 minutes. Each colour was presented for 5 minutes. The sequence of colour presentation was counterbalanced between subjects.

The dependent variable measured was the distance (cm) covered by each subject toward the red and the blue. To estimate chicks' colour preference, we calculated an index using the formula:

$$\text{Colour preference} = \frac{\text{cm toward red}}{\text{cm toward red} + \text{cm toward blue}}$$

This index could range between 0 (absolute preference for the blue) and 1 (absolute preference for the red), whereas 0.5 represented the absence of preference between the two colours.

To estimate differences between the two imprinting groups we employed a two-tailed independent samples *t*-test. To estimate colour preference, we employed one-sample two-tailed *t*-test against chance (0.5).

2. The manuscript describes the MRI protocol and recording method for a filial imprinting experiment in chicks. However, it failed to provide sufficient evidence to support the existing theories.

We would be really grateful to the Reviewer if she/he could mention the evidence he believes is needed and the existing theories she/he believes we missed to support. We will be happy to answer specifically to the Reviewer and update the manuscript accordingly.

3. The discussed results seem to be rather simplistic. The authors did not observe any significant differences between imprinted and nonimprinted (Figure 2A). This suggests a possibility of either a failed experiment or an incorrect protocol being followed for imprinting. Imprinted memories are often subtle and only a small set of neurons are modified in the brain. Therefore, MRI may not be able to identify changes in such a low subset of neurons.

We apologize for any confusion that may have arisen. We initially hypothesized that the contrast ($\text{Red}_{\text{Imp}} + \text{Blue}_{\text{Imp}} > \text{Blue}_{\text{Cont}} + \text{Red}_{\text{Cont}}$) would reveal networks associated with memory retrieval. This contrast combined the imprinting colours (red for red imprinted chicks and blue presentation for blue imprinted chicks) and was subtracted from the control colours (blue for red imprinted chicks and red for blue imprinted chicks), to illustrate the general effect of imprinting. However, this contrast did not yield any significantly activated cluster in the chick brain. Upon closer examination, we realized that the activation patterns for both contrasts, $\text{Red}_{\text{Imp}} + \text{Blue}_{\text{Imp}} > \text{baseline}$ and $\text{Blue}_{\text{Cont}} + \text{Red}_{\text{Cont}} > \text{baseline}$ were remarkably similar (refer to Figure 2A). This suggested to us that the introduction of the highly preferred red color in the Blue group acted as an imprinting stimulus. The brain's response to the red color in the Blue group closely resembled that of the Red group, indicating the onset of new imprinting. To discern whether this was secondary imprinting or a failure of blue imprinting, we conducted separate analyses for the first 10 (please see the response for Reviewer #2 for more details) and the last 20 minutes. The results revealed a blue imprinting effect in the Blue group during the

first 10 min (Figure R1, see also the response to comments #1 of Editor for more details), with a new imprinting occurring after the introduction of the red color during the test session.

In summary, significant differences were observed between imprinted and non-imprinted colors (see Figure 4 for the last 20 minutes of the Blue group, Figure 5 for the Red group, and Figure S3 for the first 10 minutes of the Blue group). Notably, our behavioral data align with fMRI findings: the behavioral experiment revealed a strong preference for red in Red imprinted chicks but no preference for blue in Blue imprinted chicks.

Although, brain changes related to imprinting memory could be subtle, in the present experiment we were able to find significant changes for imprinting memory retrieval and long-term storage in the red imprinted group. We also managed to find significant brain changes related to the onset of imprinting learning in the blue group exposed to red (please see the response for Reviewer #2 for more details).

4. The manuscript states that the authors found no significant differences between the red and blue imported groups in regard to the red stimulus (line 176). However, the “idea” to support this failed experimental protocol is incorrect. The argument that “the blue-imprinted chicks exposed to the preferred colour red immediately started a process of secondary imprinting towards red colour” has no evidence and is not in line with the established protocol. All imprinting experiments test the effect of memory alterations in adults after conditioning at the early stages of development. The protocol followed in the manuscript appears to have carried out imprinting and testing on the chicks themselves. We reiterate (see responses to Reviewer #2) that we were able to document also imprinting on blue color in the initial 10 min. Note also that filial imprinting experiments in chicks (*Gallus gallus*) usually test memory changes triggered during the first days of life not adults (please

see just some examples here (Bolhuis (1991), Regolin and Vallortigara (1995), Vallortigara et al., (1998), Lemaire et al. (2021), Miura and Matsushima (2016), Bolhuis (1999)). For more details please refer to response to Reviewer #2.

References

Bolhuis, J. J. (1991). Mechanisms of avian imprinting: a review. *Biological Reviews*, 66(4), 303-345.

Regolin, L., & Vallortigara, G. (1995). Perception of partly occluded objects by young chicks. *Perception & psychophysics*, 57, 971-976.

Vallortigara, G., Regolin, L., Rigoni, M., & Zanforlin, M. (1998). Delayed search for a concealed imprinted object in the domestic chick. *Animal Cognition*, 1, 17-24.

Lemaire, B. S., Rugani, R., Regolin, L., & Vallortigara, G. (2021). Response of male and female domestic chicks to change in the number (quantity) of imprinting objects. *Learning & Behavior*, 49, 54-66.

Miura, M., & Matsushima, T. (2016). Biological motion facilitates filial imprinting. *Animal Behaviour*, 116, 171-180.

Bolhuis, J. J. (1999). Early learning and the development of filial preferences in the chick. *Behavioural brain research*, 98(2), 245-252.

5. The manuscript does not provide evidence supporting the claim that MRI-based techniques can identify connectome variations associated with memory.

We adopted the Discussion as follows to provide evidence supporting fMRI application in identifying connectome variations associated with different forms of memory:

“fMRI has been used in several studies to investigate the connection between connectome variations and memory. A recent study in songbirds highlights fMRI's role in tracking song memory development in zebra finch's post-tutorial sessions³⁵. The study revealed permanent neural activity changes in auditory perception and song learning, highlighting early sensory

experiences. Gazzaley and Nobre³⁶ explored the neural basis of working memory encoding and maintenance using fMRI. Rahm et al.³⁷ characterized the neural basis of visual working memory recognition using fMRI by varying recognition needs and similarity between probe items and memory contents. Yang et al.³⁸ proposed an enhanced connectome-based predictive modeling approach, which showed strong applicability across different cognitive processes and could predict working memory performance in healthy individuals. These findings underscore the potential of fMRI in understanding brain processes that underpin cognitive abilities.”

6. In my opinion, the way the manuscript is to confuse the readers. As a general rule, scientific papers should clearly and concisely communicate the facts they are presenting, without requiring the reader to spend an excessive amount of time deciphering what is being said.

We are sorry about that. We did our best to make the manuscript as clear as possible in the revised version.